# StruDiCO: Structured Denoising Diffusion with Gradient-free Inference-stage Boosting for Memory and Time Efficient Combinatorial Optimization

**Yu Wang[1], Yang Li[2], Junchi Yan[2*], Yi Chang[1,3,4*]**
[1]School of Artificial Intelligence, Jilin University
[2]School of Artificial Intelligence, Shanghai Jiao Tong University
[3]International Center of Future Science, Jilin University
[4]Engineering Research Center of Knowledge-Driven Human-Machine Intelligence, MOE, China
{yu_w,yichang}@jlu.edu.cn    {yanglily,yanjunchi}@sjtu.edu.cn

## Abstract

Diffusion models have recently emerged as powerful neural solvers for combinatorial optimization (CO). However, existing approaches fail to reveal how variables are progressively determined during inference, making the final solution opaque until the last step. To address this limitation, we propose a structured denoising diffusion model, StruDiCO, which incrementally constructs solutions through step-wise variable selection. This is achieved via a variable-absorption noising model, wherein the forward process simulates gradual variable deactivation, converging to an empty solution, while the reverse process incrementally selects variables to reconstruct the final solution. This design induces structural continuity across intermediate states, enabling interpretable and trajectory-consistent partial solutions throughout inference. To further improve the reliability of reverse inference, we introduce a constrained consistency sampling strategy, which suppresses low-confidence variable selection at each step to stabilize the reverse process. Leveraging the structure-preserving reverse process, we further propose a lightweight, gradient-free, objective-aware refinement framework, which iteratively improves solution quality by applying structure-aware perturbations to the current solution, performing reverse inference through the constraint consistency model, and decoding with an objective-guided scoring scheme. Extensive experiments on two canonical CO tasks, the Traveling Salesman Problem (TSP) and Maximal Independent Set (MIS), show that StruDiCO outperforms state-of-the-art diffusion-based solvers, achieving up to $3.5\times$ faster inference, 70% lower GPU memory usage, and significantly improved solution quality, with up to 37.7% drop reduction on TSP and an average 38.1% improvement on MIS. The codes are publicly available at `https://github.com/yuuuuwang/StruDiCO`.

## 1 Introduction

Combinatorial optimization (CO) entails optimizing discrete decision variables toward specified objectives and serves as a cornerstone for numerous practical applications requiring efficient decision-making [1, 2, 3]. However, the inherent NP-hardness of CO problems renders large-scale CO instances particularly challenging to solve efficiently, traditionally necessitating the design of handcrafted

* Corresponding authors: Junchi Yan and Yi Chang (yanjunchi@sjtu.edu.cn; yichang@jlu.edu.cn). This work was partly supported by MOST of China (Grant 2023YFF0905400), NSFC (Grant U2341229, 92370201), and the China Postdoctoral Science Foundation Fellowship (Grant GZC20230947).

39th Conference on Neural Information Processing Systems (NeurIPS 2025).

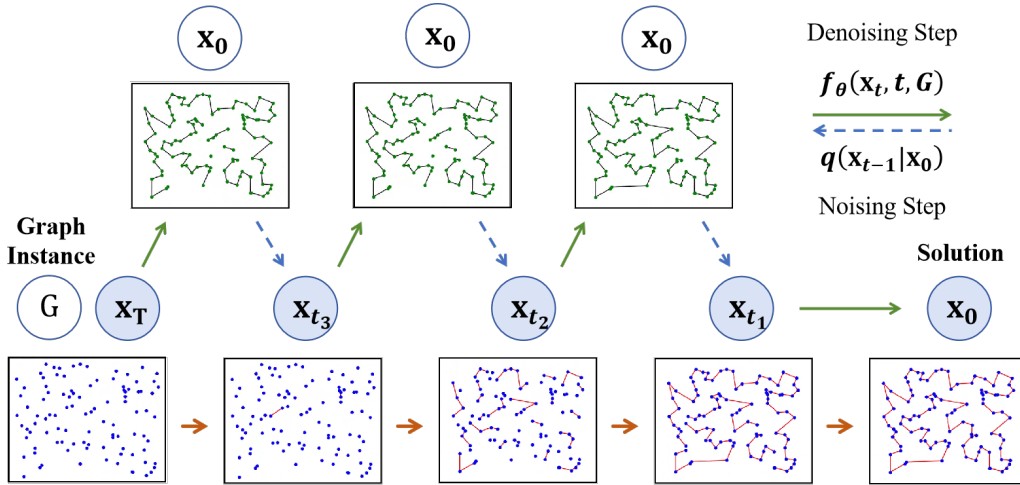

Figure 1: Overview of the structured denoising diffusion framework for combinatorial optimization. *Structured* denotes the preservation and progressive evolution of solution structures across timesteps. The forward process $(0 \rightarrow T)$ gradually removes selected edges through a variable-absorption noising model, while the reverse process incrementally reconstructs the solution by selecting edges.

heuristics. Recently, neural combinatorial optimization (NCO), including both direct solvers [4, 5, 6, 7, 8] and methods that support solving [9, 10, 11], has emerged as a data-driven paradigm, offering notable improvements in both solution quality and computational efficiency.

Existing NCO solvers can be broadly categorized based on their solution construction paradigm: autoregressive and non-autoregressive. **Autoregressive constructive solvers** [12, 13, 14, 15, 16], formulates the task as an $n$-step sequential variable selection problem, where at each step the model produces a prediction heatmap over variables, conditioning on the current partial solution to decide the next inclusion. While this step-wise generation enables fine-grained control and interpretability, it incurs significant computational overhead due to its inherently sequential nature, limiting scalability to larger instances. In contrast, **non-autoregressive solvers** [17, 18, 19, 20, 21] adopt a one-shot generation strategy. These models directly generate neural heatmaps indicating the predicted likelihood of each variable being part of the solution, followed by heuristic decoding to enforce feasibility. Among them, generative models, such as diffusion-based models [22, 23, 24] have shown strong performance by modeling instance-conditioned distribution over high-quality solutions through a sequential denoising process. However, these models still suffer from *opaque intermediate states*, providing little insight into how the final solution is gradually formed-since decisions are not explicitly revealed until the final step. **In summary**, autoregressive models provide step-wise interpretability but suffer from sequential computational overhead, whereas non-autoregressive generative models offer efficient global modeling but lack transparency in the solution trajectory. Bridging this gap is essential for developing NCO solvers that are both interpretable and efficient.

To bridge the gap between step-wise interpretability and global distribution modeling, we propose a structured discrete diffusion framework tailored for CO. Specifically, we introduce a *variable-absorption noising model*, which progressively deactivates variables during the forward process, ensuring that each intermediate state corresponds to a structurally valid partial solution. Rather than treating such interpretability as an end in itself, this structure-preserving design enables more informed and reliable decision-making throughout the inference trajectory. To enhance the reliability of reverse inference, we introduce a *constrained consistency sampling* strategy that masks low-confidence variables at each step. By narrowing the effective search space, this strategy improves structural continuity across intermediate states and stabilizes the solution trajectory. Building on this structure-preserving foundation, we propose a *lightweight gradient-free, objective-aware refinement framework* that directly incorporates task objectives into the decoding process. Unlike prior refinement methods [23, 24] that rely on gradient-based search over relaxed continuous objectives, our approach eliminates the need for backward computation entirely. The refinement proceeds in three stages: it first perturbs a feasible solution using structure-aware noise, then performs reverse inference via the consistency model, and finally reconstructs a new solution through objective-guided greedy decoding. Together, these innovations yield a diffusion-based solver that achieves high-quality solutions with

significantly reduced computational overhead, while preserving structural interpretability and aligning inference with task objectives, effectively combining the complementary strengths of autoregressive and non-autoregressive paradigms. **Our contributions toward building an interpretable and efficient neural solver for CO are summarized as follows:**

- We propose a structure-preserving discrete diffusion framework and develop StruDiCO, which leverages variable-absorption forward diffusion and constrained reverse sampling to produce interpretable, high-quality solutions for combinatorial optimization tasks.
- We introduce a gradient-free, objective-aware refinement framework that efficiently incorporates task objectives without requiring backward gradient computation [23, 24].
- We evaluate our method on TSP and MIS tasks, demonstrating superior solution quality, $3.5\times$ faster inference, and 70% lower memory usage compared to state-of-the-art neural solvers.

## 2 Preliminaries, Problem Definition and Related Works

### 2.1 Combinatorial Optimization on Graphs

Following the notations adopted in [22, 23, 24], we define $\mathcal{G}$ as the universe of CO problem instances represented by graphs $G(V, E) \in \mathcal{G}$, where $V$ and $E$ denote the node set and edge set respectively. CO problems can be broadly classified into two types based on the solution composition: edge-selecting problems which involve selecting a subset of edges, and node-selecting problems which select a subset of nodes, both subject to the feasibility constraints of the problem. Let $\mathbf{x} \in \{0, 1\}^N$ denote the optimization variable. For edge-selecting problems, $N = n^2$ and $\mathbf{x}_{i \cdot n + j}$ indicates whether $E_{ij}$ is included in $\mathbf{x}$. For node-selecting problems, $N = n$ and $\mathbf{x}_i$ indicates whether $V_i$ is included in $\mathbf{x}$. The feasible set $\Omega$ consists of $\mathbf{x}$ satisfying specific constraints as feasible solutions. A CO problem on $G$ aims to find a feasible $\mathbf{x}$ that minimize the given objective function $l(\cdot; G) : \{0, 1\}^N \to R_{\geq 0}$ :

$$\min_{\mathbf{x} \in \{0,1\}^N} l(\mathbf{x}; G) \text{ s.t. } \mathbf{x} \in \Omega$$

In this paper, we study two primary and representative CO problems: TSP and MIS. TSP defines on an undirected complete graph $G = (V, E)$, where $V$ represents $n$ cities and each edge $E_{ij}$ has a non-negative weight $w_{ij}$ representing the distance between cities $i$ and $j$. The problem can then be formulated as finding a Hamiltonian cycle of minimum weight in $G$. For MIS, given an undirected graph $G = (V, E)$, an independent set is a subset of vertices $S \subset V$ such that no two vertices in $S$ are adjacent in $G$. MIS is the problem of finding an independent set of maximum cardinality in $G$.

### 2.2 Optimization Consistency

Given a problem instance $G$ and its corresponding optimal solution $\mathbf{x}^*$, the discrete diffusion noising process defines a trajectory $\mathbf{x}_{0:T} = \mathbf{x}_0, \mathbf{x}_1, \ldots, \mathbf{x}_T$, where $\mathbf{x}_0 = \mathbf{x}^*$ and each $\mathbf{x}_t$ is sampled from the conditional distribution $q(\mathbf{x}_t \mid \mathbf{x}_0)$. In standard diffusion models, generating samples requires modeling the reverse transitions $p_\theta(\mathbf{x}_{t-1} \mid \mathbf{x}_t, G)$ through multiple iterative steps, which can be computationally expensive. To enable efficient inference, *Consistency Models* [25] introduce the notion of self-consistency, where the model learns to map any noised state $\mathbf{x}_t$ at time $t$ directly back to the clean source $\mathbf{x}_0$. Building on this idea, Fast T2T [24] gives the notion of *optimization consistency* for CO, in which all states along a noising trajectory are mapped to the same optimal solution $\mathbf{x}^*$ conditioned on the instance $G$. Formally, the consistency function $f_\theta(\mathbf{x}_t, t, G)$, parameterized by $\theta$, maps a noisy input to a denoised prediction. Under self-consistency, the training objective is to minimize the discrepancy between any two mappings along the same trajectory:

$$\mathcal{L}(\theta) = \mathbb{E}_{t_1, t_2} \left[ d \left( f_\theta(\mathbf{x}_{t_1}, t_1, G), \ f_\theta(\mathbf{x}_{t_2}, t_2, G) \right) \right], \tag{1}$$

where $d(\cdot, \cdot)$ is a distance function, e.g., binary cross-entropy. To better leverage the supervision from $\mathbf{x}^*$, an alternative minimizes the distance between each mapping and the ground-truth:

$$\mathcal{L}(\theta) = \mathbb{E}_t \left[ d \left( f_\theta(\mathbf{x}_t, t, G), \ \mathbf{x}^* \right) \right]. \tag{2}$$

This formulation can be justified by the triangle inequality, which upper-bounds the original objective:

$$\mathcal{L}(\theta) \leq \mathbb{E}_{t_1, t_2} \left[ d \left( f_\theta(\mathbf{x}_{t_1}, t_1, G), \ \mathbf{x}^* \right) + d \left( f_\theta(\mathbf{x}_{t_2}, t_2, G), \ \mathbf{x}^* \right) \right]. \tag{3}$$

This reformulation reduces the problem to supervised learning of the mapping $f_\theta$ using a standard loss, such as binary cross-entropy, and allows the model to generalize across different noise levels while consistently recovering the optimal solution.

## 2.3 Related Work

Neural combinatorial optimization (NCO) has emerged as a promising alternative to traditional heuristics such as LKH [26, 27]. Existing neural solvers can be broadly categorized into autoregressive (AR) and non-autoregressive (NAR) paradigms. An illustrative comparison with our structured diffusion approach is provided in Appendix A, highlighting key differences in solution construction.

AR methods sequentially construct solutions by selecting decision variables step-by-step. Representative works include the Attention Model (AM) [12], which first applied transformer-based architectures to routing problems, and POMO [13], which leverages policy optimization with multiple optima for reinforcement learning. Sym-NCO [14] exploits problem symmetry to enhance generalization capability, while BQ-NCO [15] formulates CO problems under a bisimulation-quotient MDP to improve robustness. The RL4CO repository [16] systematically benchmarks AR-based methods.

NAR methods aim to predict soft-constrained solutions in a one-shot manner, followed by post-processing to ensure feasibility. Early works such as GCN [17] and Att-GCN [18] employ graph neural networks [28, 29, 30] for edge prediction under **supervised learning**. UTSP [20] adopts an **unsupervised learning** framework with scattering attention GNNs, and DIMES [19] proposes a **meta-reinforcement learning** strategy combined with active search. Recently, generative models [31, 32] have shown promise in improving prediction [33], and for CO, **graph diffusion-based generative approaches** [22, 23, 24, 34] model instance-conditioned distributions over high-quality solutions. DIFUSCO [22] achieves SOTA performance on TSP and MIS but lacks instance-specific search. T2T [23] addresses this via an objective-guided gradient search, and Fast T2T [24] further applies consistency training to accelerate diffusion sampling and advance NCO performance. Unify ML4TSP [35] advances a unified modular streamline incorporating existing technologies in both learning and search for both AR and NAR methods.

# 3 Methodology

We propose a structured diffusion framework for combinatorial optimization. The process begins with a *variable-absorption noising model*, which defines a structure-preserving forward process and its reverse counterpart. To guide the reverse trajectory toward coherent solutions, we introduce a *constrained consistency sampling* strategy that selectively activates high-confidence variables at each step. Finally, we refine the decoded solutions through an *objective-aware, gradient-free method* that integrates task objectives directly into decoding without backward computation. Together, these components form a lightweight and interpretable neural solver.

## 3.1 Variable-Absorption Noising Process

Our objective is to design a forward diffusion process that generates noisy graph states while strictly respecting combinatorial constraints, e.g. preserving the validity of partial solutions throughout the noising trajectory[*]. We adopt a discrete diffusion model [22, 23, 24], tailored for structured combinatorial problems where solutions are encoded as binary vectors $\mathbf{x} \in \{0, 1\}^N$. This formulation naturally applies to edge-selection tasks such as TSP and node-selection tasks such as MIS.

It begins with an initial state $\mathbf{x}_0 \sim q(\mathbf{x}_0 \mid G)$ sampled from a data-dependent distribution that yields a feasible solution. Then, a sequence of latent states $\mathbf{x}_{1:T} = \mathbf{x}_1, \mathbf{x}_2, \ldots, \mathbf{x}_T$ is generated by progressively injecting noise while maintaining structural validity. The forward process factorizes as $q(\mathbf{x}_{1:T} \mid \mathbf{x}_0) = \prod_{t=1}^{T} q(\mathbf{x}_t \mid \mathbf{x}_{t-1})$, where each $q(\mathbf{x}_t \mid \mathbf{x}_{t-1})$ is a categorical distribution over binary states. Using a one-hot encoding $\tilde{\mathbf{x}} \in \{0, 1\}^{N \times 2}$, where each row denotes the selection status of a variable, the transition is defined as:

$$q(\mathbf{x}_t \mid \mathbf{x}_{t-1}) = \text{Cat}(\mathbf{x}_t; , \mathbf{p} = \tilde{\mathbf{x}}_{t-1}\mathbf{Q}_t), \quad q(\mathbf{x}_t \mid \mathbf{x}_0) = \text{Cat}(\mathbf{x}_t; , \mathbf{p} = \tilde{\mathbf{x}}_0\overline{\mathbf{Q}}_t), \tag{4}$$

---

[*]Since the process begins from a feasible solution $\mathbf{x}_0$ and injects noise solely by deactivating selected variables, each $\mathbf{x}_t$ along the trajectory is guaranteed to be structurally valid. This guarantee holds only for the forward process; feasibility in the reverse process typically requires post-processing.

where $\mathbf{Q}_t \in [0,1]^{2\times 2}$ is the forward transition matrix at time $t$, and $\overline{\mathbf{Q}}_t = \mathbf{Q}_1\mathbf{Q}_2\cdots\mathbf{Q}_t$ is the cumulative transition up to time $t$. In prior works [22, 23, 24], the transition matrix $\mathbf{Q}_t$ is typically symmetric and doubly stochastic: $\mathbf{Q}_t = \begin{bmatrix} 1-\beta_t & \beta_t \\ \beta_t & 1-\beta_t \end{bmatrix}$, where $\beta_t \in [0,1]$ denotes the transition rate at step $t$. This formulation enables reversible transitions and yields a uniform stationary distribution, but does not guarantee structural validity of intermediate states.

**Variable-Absorption Noising Model.** To enforce structural validity at all timesteps, we propose a *variable-absorption noising model* that induces a monotonic decay in variable activations. Specifically, a variable can remain unselected ($\mathbf{x}_i = 0$) or transition from selected ($\mathbf{x}_i = 1$) to unselected, but cannot revert once dropped. This yields a valid absorption trajectory from $\mathbf{x}_0$ to the null solution $\mathbf{x}_T = \mathbf{0}_N \in 0^N$, in which each intermediate state is a feasible partial solution. We implement this behavior using the following asymmetric transition matrix: $\mathbf{Q}_t = \begin{bmatrix} 1 & 0 \\ \beta_t & 1-\beta_t \end{bmatrix}$, where $\beta_t \in [0,1]$ denotes the absorption rate at step $t$. Under this formulation, unselected variables are frozen, and selected variables are independently dropped with probability $\beta_t$. This directional and irreversible transition enforces monotonic variable deactivation and ensures that combinatorial constraints are respected throughout the forward trajectory.

## 3.2 Constrained Consistency Sampling for Reverse Optimization

Following [24, 25], we adopt a multi-step consistency sampling procedure to progressively construct the reverse trajectory of solutions. The process begins from the empty solution $\mathbf{x}_T = \mathbf{0}_N$, and at each timestep $\tau_n$, we aim to generate a more refined intermediate state $\mathbf{x}_{\tau_n}$. Accordingly, we use a consistency model $f_\theta$ to predict a clean solution $\mathbf{x}_0$ from the current noisy input: $p_\theta(\mathbf{x}_0 \mid \mathbf{x}_{\tau_n}, \tau_n, G) = f_\theta(\mathbf{x}_{\tau_n}, \tau_n, G)$, $\mathbf{x}_0 \sim$ Bernoulli($p_\theta(\mathbf{x}_0 = 1 \mid G)$). This predicted $\mathbf{x}_0$ serves as a high-confidence proxy, guiding the sampling of the next intermediate state $\mathbf{x}_{\tau_{n+1}}$ via the variable-absorption process: $\mathbf{x}_{\tau_{n+1}} \sim$ Cat($\mathbf{p} = \tilde{\mathbf{x}}_0 \overline{\mathbf{Q}}_{\tau_{n+1}}$), where $\overline{\mathbf{Q}}_\tau$ denotes

---

**Algorithm 1** Multistep Constrained Consistency Sampling

**Require:** Consistency model $f_\theta(\cdot, \cdot, \cdot)$, graph instance $G$, time sequence $\tau_1 > \cdots > \tau_{N_\tau-1}$, threshold $\delta$
1: $\mathbf{x}_T = \mathbf{0}_N$
2: $p_\theta(\mathbf{x}_0 \mid G) \leftarrow f_\theta(\mathbf{x}_T, T, G)$
3: $\mathbf{x}_0 \sim p_\theta(\mathbf{x}_0 \mid G)$
4: **for** $n = 1$ to $N_\tau - 1$ **do**
5:     // variable-absorption process
6:     $\mathbf{x}_{\tau_n} \sim$ Cat($\mathbf{p} = \tilde{\mathbf{x}}_0 \overline{\mathbf{Q}}_{\tau_n}$)
7:     $p_\theta(\mathbf{x}_0 \mid G) \leftarrow f_\theta(\mathbf{x}_{\tau_n}, \tau_n, G)$
8:     // constrained sampling over $\mathbf{m}$
9:     $\mathbf{m} \leftarrow \mathbb{I}(p_\theta(\mathbf{x}_0 = 1 \mid G) > \delta)$
10:    $\mathbf{x}_0 \sim$ Bernoulli($\mathbf{m} \odot p_\theta(\mathbf{x}_0 = 1 \mid G)$)
11: **end for**
12: **return** Solution $\mathbf{x}_0$

---

the cumulative absorption transition matrix. The key modeling goal is to construct a trajectory of intermediate states $\mathbf{x}_T, \mathbf{x}_{\tau_1}, \ldots, \mathbf{x}_{N_\tau-1}, \mathbf{x}_0$ that exhibit increasing structure and semantic coherence, ultimately leading to a high-quality solution. This construction process alternates between denoising (via $f_\theta$) and noise injection (via variable-absorption), forming a progressive trajectory over $\mathbf{x}_\tau$. Although each $\mathbf{x}_0$ prediction is discarded after use, it plays a crucial role in steering the generation of the next $\mathbf{x}_\tau$. As illustrated in Fig. 1, this reverse process forms a zigzag pattern between model predictions $\mathbf{x}_0$ and intermediate states $\mathbf{x}_\tau$ with increasing structural coherence, progressively guiding the construction of the final solution. It is important to note that structural coherence across reverse steps is primarily induced by the variable-absorption process, which softly enforces a monotonic selection trajectory ($\mathbf{x}_{\tau_n} \subseteq \mathbf{x}_{\tau_{n+1}}$). For stricter structural guarantees, see our discussion in Appendix B.

**Constrained Sampling.** To reduce noise and improve the stability of this process, we introduce a *threshold-based constrained sampling strategy*. After obtaining the prediction $p_\theta(\mathbf{x}_0 = 1 \mid G)$, we suppress low-confidence variables by applying a hard threshold $\delta$:

$$\mathbf{m} = \mathbb{I}(p_\theta(\mathbf{x}_0 = 1 \mid G) > \delta), \quad \mathbf{x}_0 \sim \text{Bernoulli}(\mathbf{m} \odot p_\theta(\mathbf{x}_0 = 1 \mid G)). \quad (5)$$

The constrained sampling strategy is also supported by theoretical insights. Combinatorially, it reduces the candidate solution space from $2^N$ to at most $\sum_{k=0}^{\gamma N} \binom{N}{k}$ for a thresholded fraction $\gamma \in (0,1)$. Statistically, thresholding the predicted heatmap can be seen as truncating the predictive distribution to a high-confidence region, reducing entropy and sample variance. See proof in Appendix C.

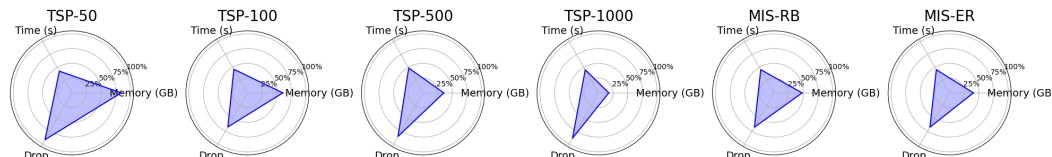

Figure 2: Relative comparison of StruDiCO and Fast T2T across six combinatorial optimization tasks. Metrics include GPU memory usage, inference time, and performance drop (all normalized to Fast T2T = 100%). Lower values indicate improved efficiency or solution quality.

Table 1: Summary of objective-guided refinement framework. StruDiCO enables multi-step *structure construction*, where intermediate states are valid subsets of the soft-constrained solution. $B$: batch size; $L$: number of layers; $N$: number of nodes; $d$: hidden dimension.

| Method | Multi-Step Structure Construction | Objective-Guided | | | Complexity | |
| --- | --- | --- | --- | --- | --- | --- |
| | | Gradient-free | Forward | Backward | Time | Memory |
| Fast T2T [24] | ✗ | ✗ | 2 | 1 | $\sim O(3BLN^2d)$ | $\sim O(2BLN^2d)$ |
| StruDiCO (Ours) | ✓ | ✓ | 1 | 0 | $O(BLN^2d)$ | $O(BLN^2d)$ |

**Theorem 3.1** (Constrained Sampling Reduces Divergence). *Let $q(\mathbf{x}_0 \mid G)$ denote the true distribution and $p_\theta(\mathbf{x}_0 \mid \mathbf{x}_\tau, \tau, G)$ the model prediction. Let $\tilde{p}_\theta = Mask_\delta(p_\theta)$ be the renormalized distribution obtained by thresholding low-confidence variables, $\tilde{p}_\theta(\mathbf{x}_0) = \frac{\mathbb{I}[p_\theta(\mathbf{x}_0) > \delta] \, p_\theta(\mathbf{x}_0)}{Z_\delta}$, $Z_\delta = \sum_{\mathbf{x}_0} \mathbb{I}[p_\theta(\mathbf{x}_0) > \delta] \, p_\theta(\mathbf{x}_0)$, where $Z_\delta$ is the normalization constant representing the retained probability mass. Then, in terms of the reverse KL divergence, we have $D_{\mathrm{KL}}(\tilde{p}_\theta \| q) \leq D_{\mathrm{KL}}(p_\theta \| q) + \varepsilon_\delta$, where the slack term $\varepsilon_\delta$ vanishes as the discarded probability mass $1 - Z_\delta \to 0$.*

**Remark.** Theorem 3.1 formally shows that threshold-based constrained sampling reduces the divergence between the model and the true solution distributions. Specifically, by masking out low-confidence regions (probability $< \delta$) and renormalizing the remaining mass, the KL divergence decreases whenever the retained region has higher expected likelihood than the discarded region, a condition that is naturally satisfied during inference where high-confidence variables dominate. Importantly, while constrained sampling can also be applied to uniform-noise diffusion models [22, 24], its benefits are substantially amplified under the structure-preserving variable-absorption process, which provides a monotonic and semantically consistent reverse trajectory (see Sec. 4.3).

### 3.3 Objective-Aware Gradient-Free Refinement

Following the reverse sampling process in Algorithm 1, we finally sample a soft-constrained solution $\mathbf{x}_0 \sim p_\theta(\mathbf{x}_0 \mid G)$. To obtain a feasible solution $\mathbf{s} \in \Omega$, we apply greedy decoding by sequentially inserting edges (for TSP) or nodes (for MIS) with the highest confidence if there are no conflicts, enforcing task-specific constraints. This step ensures hard feasibility and is commonly used in [19, 22, 23, 24]. While feasible, it does not explicitly optimize the task objective and often yields suboptimal solutions. Recent methods like T2T [23] and Fast T2T [24] address this by incorporating task-level supervision via gradient-guided refinement. These methods itera-

---

**Algorithm 2** Objective-Aware Gradient-Free Refinement

**Require:** Consistency model $f_\theta$, graph instance $G$, prediction $p_\theta$, score function $\text{score}_i = p_i / \phi_i$, number of iterations $T_g$, perturbation ratio $\alpha T$
1: **for** $t = 1$ to $T_g$ **do**
2: $\quad \mathbf{m} \leftarrow \mathbb{I}(p_\theta(\mathbf{x}_0 = 1 \mid G) > \delta)$
3: $\quad \mathbf{s} \leftarrow \mathbf{m} \odot \mathbf{s}$
4: $\quad \tilde{\mathbf{s}} \leftarrow \text{ONEHOTENCODE}(\mathbf{s})$
5: $\quad$ //apply variable-absorption noise
6: $\quad \mathbf{s}_{\alpha T} \sim \text{Cat}(\tilde{\mathbf{s}} \cdot \overline{\mathbf{Q}}_{\alpha T})$
7: $\quad p_\theta \leftarrow f_\theta(\mathbf{s}_{\alpha T} \mid G)$
8: $\quad \mathbf{s} \leftarrow \text{GREEDYDECODE}(p_\theta, \text{score})$
9: **end for**
10: **return** $\mathbf{s}$

---

tively perturb a solution and use reverse denoising guided by objective gradients to refine solution, but at the cost of repeated forward-backward passes and reliance on differentiable approximations, making them computationally expensive and less suited to discrete combinatorial settings.

To overcome these practical inefficiencies, we propose a lightweight *objective-aware gradient-free refinement* strategy. Rather than backpropagating gradients, we integrate task objectives directly into

Table 2: Results on TSP-50 and TSP-100. G: Greedy Decoding.

| Algorithm | Type | TSP-50 | | | TSP-100 | | |
|---|---|---|---|---|---|---|---|
| | | Length↓ | Drop↓ | Time↓ | Length↓ | Drop↓ | Time↓ |
| Concorde [36] | Exact | 5.688 | 0.00% | 0.074s | 7.756 | 0.00% | 0.404s |
| LKH3 [26] | Heuristics | 5.688 | 0.00% | 0.058s | 7.756 | 0.00% | 0.176s |
| GCN [17] | SL+G+2Opt | 5.694 | 0.115% | 0.009s | 7.807 | 0.649% | 0.019s |
| GNNGLS* [37] | SL+G+2Opt | 5.707 | 0.333% | 0.019s | 7.857 | 1.295% | 0.129s |
| DIMES* [19] | Meta+RL+G+2Opt | 5.823 | 2.387% | 0.018s | 8.007 | 3.232% | 0.057s |
| AM* [12] | RL+G+2Opt | 5.679 | 0.167% | 0.048s | 7.826 | 0.898% | 0.438s |
| POMO [13] | RL+G+2Opt | 5.693 | 0.102% | 0.019s | 7.854 | 1.253% | 0.116s |
| Sym-NCO [14] | RL+G+2Opt | 5.694 | 0.122% | 0.198s | 7.818 | 0.796% | 0.634s |
| BQ-NCO* [15] | RL+G+2Opt | 5.795 | 1.894% | 0.205s | 7.893 | 1.772% | 0.387s |
| DIFUSCO ($T_s$=50) [22] | SL+G | 5.692 | 0.076% | 0.229s | 7.851 | 1.216% | 0.591s |
| Fast T2T ($T_s$=3) [24] | SL+G | 5.694 | 0.111% | 0.024s | 7.798 | 0.537% | 0.038s |
| StruDiCO ($T_s$=3) | SL+G | 5.692 | 0.071% | 0.023s | 7.786 | 0.392% | 0.037s |
| T2T ($T_s$=50,$T_g$=30) [23] | SL+G | 5.688 | 0.015% | 0.717s | 7.765 | 0.125% | 1.559s |
| Fast T2T ($T_s$=3,$T_g$=3) [24] | SL+G | 5.688 | 0.014% | 0.139s | 7.760 | 0.052% | 0.182s |
| StruDiCO ($T_s$=3,$T_g$=3) | SL+G | 5.688 | 0.014% | 0.051s | 7.758 | 0.036% | 0.082s |
| DIFUSCO ($T_s$=50) [22] | SL+G+2Opt | 5.690 | 0.046% | 0.23s | 7.776 | 0.262% | 0.590s |
| Fast T2T ($T_s$=3) [24] | SL+G+2Opt | 5.689 | 0.031% | 0.026s | 7.764 | 0.101% | 0.037s |
| StruDiCO ($T_s$=3) | SL+G+2Opt | 5.689 | 0.019% | 0.026s | 7.761 | 0.067% | 0.038s |
| T2T ($T_s$=50,$T_g$=30) [23] | SL+G+2Opt | 5.688 | 0.012% | 1.098s | 7.760 | 0.058% | 1.571s |
| Fast T2T ($T_s$=3,$T_g$=3) [24] | SL+G+2Opt | 5.688 | 0.012% | 0.139s | 7.759 | 0.036% | 0.180s |
| StruDiCO ($T_s$=3,$T_g$=3) | SL+G+2Opt | 5.688 | 0.011% | 0.059s | 7.756 | 0.024% | 0.083s |

greedy decoding using a objective-aware scoring scheme. Inspired by the relaxed objective redefined in [23], we combine task rewards and constraint penalties to form guidance signals. Specifically, we define the variable-wise score as $\text{score}_i = p_i/(\phi_i + \epsilon)$, where $\epsilon$ serves as a balancing term, $p_i = p_\theta(\mathbf{x}_0^i = 1 \mid G)$ denotes the predicted confidence, and $\phi_i$ encodes task-specific penalties:

$$\phi(i)_{\text{TSP}} = d(i) \quad \text{(edge distance)}, \qquad \phi(i)_{\text{MIS}} = \sum_{j \in \mathcal{N}(i)} p_j \quad \text{(conflict potential)} \qquad (6)$$

This scoring scheme favors confident variables that contribute positively to the task objective, e.g., shorter edges in TSP or nodes likely to form larger independent sets in MIS, thus guiding decoding in an objective-aligned direction without requiring gradient computation.

**Gradient-Free Iterative Refinement.** Building on the proposed objective-aware scoring scheme, we further introduce an iterative refinement strategy to improve solution quality. Starting from a feasible solution $\mathbf{s}_0$, we apply structure-aware perturbations via variable-absorption noise, denoise the perturbed state using the consistency model, and greedily decode a new solution using the objective-aware score. This process explores a local neighborhood around the current solution and concentrates inference within high-confidence regions. The refinement is fully forward-pass and can be iterated with minimal overhead, as detailed in Algorithm 2. We justify this procedure through the following informal proposition.

**Proposition 3.1** (Refinement Enhances Solution Reliability, Informal). *Let $\mathbf{s}$ be a feasible solution decoded from a predictive distribution $p_\theta$ using an objective-aware greedy strategy. Assume that the variable-absorption noise operator $\overline{\mathbf{Q}}_\alpha$ introduces bounded perturbations that preserve the high-confidence regions of $p_\theta$, and that the consistency model $f_\theta$ is locally Lipschitz continuous under such perturbations. Then, iterative refinement converges toward fixed points that remain within locally stable, high-confidence regions of the predictive space.*

**Remark.** Our gradient-free refinement strategy is fundamentally enabled by the structural consistency of the variable-absorption noising model. Since each refinement step perturbs only a subset of already selected variables, the resulting inputs remain semantically aligned with the current solution. This local consistency is preserved by the denoising process and reinforced by objective-aware greedy decoding, enabling stable and meaningful refinement across iterations. In contrast, prior diffusion models based on uniform noise [22, 23, 24] apply symmetric noise to all variables regardless of selection status, which destroys partial solution structure and produces perturbed inputs that are poorly aligned with the original solution. As a result, consistency denoising from such noisy inputs lacks directional guidance, and greedy decoding does not provide reliable improvement. Therefore, while

Table 3: Results on TSP-500 and TSP-1000. S: Sampling Decoding.

| Algorithm | Type | TSP-500 | | | TSP-1000 | | |
|---|---|---|---|---|---|---|---|
| | | Length↓ | Drop↓ | Time | Length↓ | Drop↓ | Time |
| *Mathematical Solvers or Heuristics* | | | | | | | |
| Concorde [36] | Exact | 16.546 | 0.00% | 18.672s | 23.118 | 0.00% | 84.413s |
| LKH-3 [26] | Heuristics | 16.546 | 0.00% | 1.848s | 23.119 | 0.00% | 4.641s |
| *Learning-based Solvers with Greedy Decoding* | | | | | | | |
| GCN [17] | SL+G+2OPT | 16.899 | 2.121% | 0.128s | – | – | – |
| DIMES* [19] | RL+G+2Opt | 17.165 | 3.742% | 0.453s | – | – | – |
| BQ-NCO* [15] | RL+G+2Opt | 16.838 | 1.766% | 2.454s | 23.647 | 2.287% | 5.722s |
| DIFUSCO ($T_s$=50) [22] | SL+G | 18.136 | 9.611% | 1.442s | 25.667 | 11.022% | 4.982s |
| Fast T2T ($T_s$=5) [24] | SL+G | 17.467 | 5.551% | 0.251s | 24.698 | 6.831% | 0.971s |
| StruDiCO ($T_s$=5) | SL+G | 17.404 | 5.172% | 0.239s | 23.118 | 5.967% | 0.900s |
| T2T ($T_s$=50,$T_g$=30) [23] | SL+G | 17.470 | 5.578% | 3.334s | 25.168 | 8.868% | 12.871s |
| Fast T2T ($T_s$=5,$T_g$=5) [24] | SL+G | 16.919 | 2.244% | 1.426s | 23.936 | 3.539% | 5.988s |
| StruDiCO ($T_s$=5,$T_g$=5) | SL+G | 16.887 | 2.045% | 0.651s | 23.755 | 2.753% | 2.533s |
| DIFUSCO ($T_s$=50) [22] | SL+G+2Opt | 16.817 | 1.641% | 1.433s | 23.567 | 1.936% | 5.036s |
| Fast T2T ($T_s$=5) [24] | SL+G+2Opt | 16.701 | 0.922% | 0.261s | 23.388 | 1.167% | 0.979s |
| StruDiCO ($T_s$=5) | SL+G+2Opt | 16.669 | 0.728% | 0.247s | 23.348 | 0.996% | 0.915s |
| T2T ($T_s$=50,$T_g$=30) [23] | SL+G+2Opt | 16.677 | 0.793% | 3.367s | 23.397 | 1.209% | 13.089s |
| Fast T2T ($T_s$=5,$T_g$=5) [24] | SL+G+2Opt | 16.611 | 0.383% | 1.387s | 23.257 | 0.603% | 5.779s |
| StruDiCO ($T_s$=5,$T_g$=5) | SL+G+2Opt | 16.602 | 0.326% | 0.674s | 23.242 | 0.535% | 2.614s |
| *Learning-based Solvers with 4× Sampling Decoding* | | | | | | | |
| DIFUSCO ($T_s$=50) [22] | SL+S | 17.533 | 5.959% | 2.131s | 25.059 | 8.399% | 19.023s |
| Fast T2T ($T_s$=5) [24] | SL+S | 17.024 | 2.874% | 0.877s | 24.096 | 4.231% | 3.635s |
| StruDiCO ($T_s$=5) | SL+S | 16.995 | 2.696% | 0.839s | 24.007 | 3.849% | 3.430s |
| T2T ($T_s$=50,$T_g$=30) [23] | SL+S | 17.054 | 3.070% | 9.722s | 24.838 | 7.444% | 34.813s |
| Fast T2T ($T_s$=5,$T_g$=5) [24] | SL+S | 16.710 | 0.978% | 5.216s | 23.712 | 2.570% | 13.346s |
| StruDiCO ($T_s$=5,$T_g$=5) | SL+S | 16.688 | 0.846% | 2.312s | 23.471 | 1.527% | 6.520s |
| DIFUSCO ($T_s$=50) [22] | SL+S+2Opt | 16.694 | 0.893% | 4.941s | 23.425 | 1.326% | 19.217s |
| Fast T2T ($T_s$=5) [24] | SL+S+2Opt | 16.633 | 0.509% | 0.911s | 23.286 | 0.726% | 3.733s |
| StruDiCO ($T_s$=5) | SL+S+2Opt | 16.611 | 0.383% | 0.857s | 23.273 | 0.670% | 3.541s |
| T2T ($T_s$=50,$T_g$=30) [23] | SL+S+2Opt | 16.621 | 0.453% | 12.636s | 23.371 | 1.070% | 36.271s |
| Fast T2T ($T_s$=5,$T_g$=5) [24] | SL+S+2Opt | 16.580 | 0.194% | 5.095s | 23.287 | 0.419% | 13.029s |
| StruDiCO ($T_s$=5,$T_g$=5) | SL+S+2Opt | 16.575 | 0.168% | 2.437s | 23.178 | 0.261% | 6.836s |

refinement can in principle also be applied to uniform-noise models, its effectiveness is substantially amplified under the monotonic, structure-preserving property of variable-absorption diffusion.

We summarize the key differences between StruDiCO and prior work [24] in Table 1. Unlike Fast T2T, which incorporates objective guidance during the denoising process, StruDiCO integrates task objectives directly into the decoding stage. This design eliminates the need for backward gradient computation, resulting in significantly lower time and memory complexity while enabling finer control over structure-aware refinement. In addition, Fig. 2 presents a comparative evaluation across six benchmark tasks in terms of GPU memory usage, inference time, and solution quality (Drop), with all metrics normalized to Fast T2T (100%). StruDiCO consistently achieves superior efficiency and performance, with notable improvements on large-scale instances such as TSP-1000 and MIS-ER.

# 4 Experiment

We evaluate our method on two classic combinatorial optimization tasks-Traveling Salesman Problem (TSP) and Maximum Independent Set (MIS)-across a wide range of graph sizes and difficulty levels. Our evaluation includes comparisons against state-of-the-art (SOTA) exact solvers, heuristics, and neural baselines. For fair comparison, we adopt the standard diffusion-based configuration where $T_s$ denotes the number of initial inference steps and $T_g$ the number of guided refinement steps.

We report three metrics across all settings: (1) **Objective**, the value of the solution (tour *length* for TSP, subset *size* for MIS); (2) **Drop**, the relative performance degradation compared to reference solutions; and (3) **Time**, the average runtime per instance. All models are evaluated with batch size 1 and in single-thread mode unless otherwise specified. More details are provided in the Appendix E.

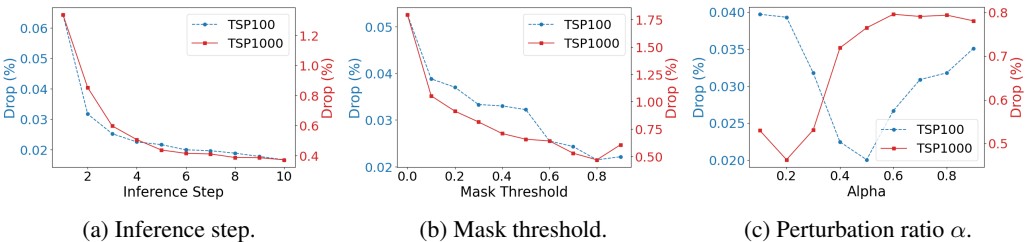

|             | (a) Inference step. | (b) Mask threshold. | (c) Perturbation ratio $\alpha$. |

Figure 3: Hyperparameter study on different aspects of our method.

## 4.1 Results on TSP

**Datasets.** We follow prior works [12, 22, 23, 24, 35] to generate TSP instances with $N \in \{50, 100, 500, 1000\}$ nodes, uniformly sampled from the unit square. A TSP instance includes $N$ 2-D coordinates and a reference solution obtained by heuristics.

**Main Results.** As shown in Tables 2 and 3, our method achieves highly competitive solution quality under both greedy and sampling decoding ($\times 4$). For example, on TSP-1000, our method achieves a relative drop of 0.535%, outperforming Fast T2T's 0.603% under the same $T_s = 5$, $T_g = 5$ configuration. In terms of runtime, our method is $2.2\times$ **faster** on average compared to Fast T2T and over $5\times$ **faster** than T2T. Moreover, Fig. 2 presents radar plots for memory, runtime, and drop across TSP variants, normalized to Fast T2T. Our method consistently reduces GPU memory by **30%–70%**, achieves **40%–65% faster** runtime, and improves solution quality, highlighting the benefit of our gradient-free refinement and constrained denoising.

**Hyperparameter Study.** We conduct hyperparameter studies on TSP-100 and TSP-1000 to analyze the impact of three core hyperparameters in our method: *(1) Inference step and guided step ($T_s = T_g$).* As shown in Fig. 3a, increasing the number of steps improves solution quality up to a point, with performance saturating around $T_s = T_g = 5$ for both instance sizes. *(2) Mask threshold ($\delta$).* Fig. 3b indicates that setting an appropriate threshold $\delta$ is critical. Low thresholds retain excessive noise, while overly high values limit exploration. The results demonstrate the tradeoff existence and indicate that $\delta = 0.8$ can achieve relatively better performance. *(3) Perturbation ratio ($\alpha$).* As shown in Fig. 3c, the refinement quality depends sensitively on the perturbation

Table 4: Generalization results with *Greedy Decoding*.

| Testing | Training | TSP-50 | TSP-100 | TSP-500 | TSP-1K |
|---|---|---|---|---|---|
| TSP-50 | DIFUSCO ($T_s$=50)* [22] | 0.09% | 0.25% | 2.55% | 2.71% |
| | T2T ($T_s$=50, $T_g$=30)* [23] | 0.02% | 0.11% | 1.60% | 1.10% |
| | Fast T2T ($T_s$=5, $T_g$=5) [24] | 0.09% | 0.36% | 1.02% | 1.26% |
| | StruDiCO ($T_s$=5, $T_g$=5) | **0.01%** | **0.01%** | **0.31%** | **0.80%** |
| TSP-100 | DIFUSCO ($T_s$=50)* [22] | 1.44% | 0.23% | 3.44% | 3.31% |
| | T2T ($T_s$=50, $T_g$=30)* [23] | 0.56% | 0.17% | 2.47% | 2.19% |
| | Fast T2T ($T_s$=5, $T_g$=5) [24] | **0.12%** | 0.02% | 0.40% | 0.55% |
| | StruDiCO ($T_s$=5, $T_g$=5) | 0.16% | **0.02%** | **0.34%** | **0.55%** |
| TSP-500 | DIFUSCO ($T_s$=50)* [22] | 4.16% | 3.04% | 1.40% | 1.85% |
| | T2T ($T_s$=50, $T_g$=30)* [23] | 3.79% | 2.25% | 0.91% | 1.22% |
| | Fast T2T ($T_s$=5, $T_g$=5) [24] | **2.67%** | **1.77%** | 0.38% | 0.95% |
| | StruDiCO ($T_s$=5, $T_g$=5) | 3.46% | 2.96% | **0.33%** | **0.49%** |
| TSP-1K | DIFUSCO ($T_s$=50)* [22] | 4.54% | 3.98% | 2.65% | 2.21% |
| | T2T ($T_s$=50, $T_g$=30)* [23] | 4.66% | 3.81% | 1.61% | 1.30% |
| | Fast T2T ($T_s$=5, $T_g$=5) [24] | **3.46%** | **3.08%** | 1.06% | 0.58% |
| | StruDiCO ($T_s$=5, $T_g$=5) | 4.31% | 3.22% | **0.75%** | **0.54%** |

ratio ($\alpha$). Too small values fail to explore alternative candidates, while too large values destroy prior structure. A moderate value ($0.2 \leq \alpha \leq 0.5$) leads to the lowest Drop for TSP-100 and TSP-1000.

**Results for Generalization.** We evaluate the cross-scale generalization ability of models by testing on TSP instances of varying sizes, while keeping the training set fixed. Table 4 reports the Drop (%) under greedy decoding. StruDiCO achieves the lowest performance drop in most settings. Notably, when tested on TSP-50, the model trained on TSP-1000 yields a drop of only 0.80%, outperforming Fast T2T (1.26%) and DIFUSCO (2.71%). Similar trends are observed on other test sizes (e.g., TSP-500 or TSP-1K), demonstrating that StruDiCO generalizes well across scales.

## 4.2 Results on MIS

**Datasets.** Following prior work [19, 22, 23, 24, 40, 41, 42], we use two types of graphs: random RB graphs [43] with 200-300 nodes and Erdős–Rényi (ER) graphs [44] with 700-800 nodes, where each edges appears independently with probability $p = 0.15$.

**Main Results.** As shown in Table 5, StruDiCO consistently outperforms diffusion-based baselines in both solution quality and efficiency. On ER graphs, it achieves the best Drop across all decoding

Table 5: Results on MIS. * quoted from [23, 24].

| Algorithm | Type | RB-[200-300] | | | ER-[700-800] | | |
|---|---|---|---|---|---|---|---|
| | | SIZE↑ | Drop↓ | Time | SIZE↑ | Drop↓ | Time |
| KaMIS [38] | Heuristics Exact | 20.090* | 0.00% | 45.809s | 44.969* | 0.00% | 60.753s |
| Gurobi [39] | Exact | 20.090 | 0.00% | 22.033s | 41.28 | 8.203% | 23.437s |
| DIFUSCO* ($T_s = 100$) [22] | SL+G | 18.52 | 7.81% | – | 36.667 | 18.462% | 4.923s |
| Fast T2T ($T_s$=5) [24] | SL+G | 18.875 | 6.023% | 0.085s | 37.820 | 15.898% | 0.262s |
| StruDiCO ($T_s$=5) | SL+G | 19.308 | 3.899% | 0.086s | 39.820 | 11.450% | 0.261s |
| T2T* ($T_s = 50, T_g = 30$) [23] | SL+G | 18.98 | 5.49% | – | 39.833 | 11.419% | 6.253s |
| Fast T2T ($T_s$=5,$T_g$=5) [24] | SL+G | 19.578 | 2.540% | 0.399s | 40.781 | 9.313% | 1.217s |
| StruDiCO ($T_s$=5,$T_g$=5) | SL+G | 19.748 | 1.707% | 0.181s | 42.125 | 6.324% | 0.552s |
| DIFUSCO* ($T_s = 100$) [22] | SL+S | 19.13 | 4.79% | – | 39.083 | 13.092% | 17.939s |
| Fast T2T ($T_s$=5) [24] | SL+S | 19.430 | 3.29% | 0.228s | 39.767 | 11.568% | 1.0456s |
| StruDiCO ($T_s$=5) | SL+S | 19.756 | 1.687% | 0.229s | 41.531 | 7.645% | 0.941s |
| T2T* ($T_s = 50, T_g = 30$) [23] | SL+S | 19.38 | 3.53% | – | 41.417 | 7.899% | 22.426s |
| Fast T2T ($T_s$=5,$T_g$=5) [24] | SL+S | 19.748 | 1.715% | 1.079s | 41.651 | 7.378% | 4.379s |
| StruDiCO ($T_s$=5,$T_g$=5) | SL+S | 19.908 | 0.934% | 0.492s | 43.265 | 3.789% | 1.988s |

Table 6: Ablation study on TSP and MIS tasks (Gap, ↓).

| Variant | TSP-50 | TSP-100 | TSP-500 | TSP-1000 | MIS-RB | MIS-ER |
|---|---|---|---|---|---|---|
| Uniform CM (Fast T2T, $T_s$=5) | 0.031% | 0.101% | 0.922% | 1.167% | 6.023% | 15.898% |
| VA CM (StruDiCO) | 0.032% | 0.163% | 1.091% | 1.183% | 7.476% | 23.461% |
| Uniform + CCS | 0.025% | 0.081% | 0.898% | 1.074% | 5.878% | 16.522% |
| VA + CCS (StruDiCO, $T_s$=5) | 0.019% | 0.067% | 0.728% | 0.797% | 3.899% | 11.450% |

modes: **6.324%** under greedy decoding and **3.789%** under sampling, compared to **9.313% / 7.378%** from Fast T2T and **11.419% / 7.899%** from T2T. In terms of runtime, StruDiCO is **11× faster** than T2T and **over 2× faster** than Fast T2T under the same refinement setup ($T_s = 5, T_g = 5$).

**Generalization.** Figures 6 - 9 show generalization performance on ER graphs with varying edge density ($p$ from 0.2 to 0.4) and graph size (from 350-400 to 1400-1600). Our method achieves consistently better solution sizes, outperforming Fast T2T by up to **8%** and DIFUSCO by over **20%** in sparse settings, while maintaining superior scalability.

## 4.3 Ablation Study

We conduct a detailed ablation study to disentangle the contributions of the Variable-Absorption (VA) mechanism and Constrained Consistency Sampling (CCS). The results are summarized in Table 6. **1)** Replacing the uniform consistency model (CM) with VA slightly degrades performance. This is because the monotonic absorption process ($1 \rightarrow 0$) produces conservative trajectories, structurally coherent but with limited exploration. By contrast, the uniform CM allows symmetric noise injection ($0 \leftrightarrow 1$), which supports broader exploration. **2)** Although CCS improves both uniform and VA models, the gain is substantially larger when combined with VA. This highlights that VA provides semantically consistent intermediate states, enabling CCS to operate in a low-noise, structured space. **Overall**, the ablation confirms that VA and CCS are complementary: VA provides structural continuity, CCS leverages it for reliable variable selection, together accounting for the strong performance.

## 5 Conclusion

We have introduced a structured denoising diffusion framework tailored for CO problems. By modeling the forward process as a variable absorption process and the reverse process as a progressive variable selection mechanism, it enables the construction of interpretable partial solutions at each inference step. We also propose a gradient-free objective-aware decoding strategy that explicitly incorporates objective constraints to enhance inference quality on unseen instances. Extensive experiments on TSP and MIS show the superior efficiency and effectiveness, achieving up to $3.5\times$ faster inference, 70% lower memory usage, and substantial improvements in solution quality.

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

# A Discussion on the Construction Paradigms of NCO

As illustrated in Fig. 4, **autoregressive** (AR) methods construct solutions by selecting decision variables (nodes or edges) in a strictly sequential manner, typically requiring $O(N)$ decoding steps, where $N$ is the number of decision variables. In contrast, **non-autoregressive** (NAR) methods predict soft-constrained solutions in a one-shot pass, followed by heuristic post-processing to ensure feasibility, effectively reducing decision-making to a single step. These two paradigms represent the two extremes of variable selection granularity in neural combinatorial optimization (NCO).

Naturally, this raises the question: *Can we design a construction paradigm that balances granularity and efficiency, lying between these two extremes?* Recently, generative approaches based on diffusion models have achieved state-of-the-art performance and offer a promising direction. However, they suffer from two critical limitations:

1. **Excessive inference steps.** Traditional diffusion models typically involve hundreds or thousands of steps (e.g., $T = 1000$), often exceeding the number of variables in the problem, which introduces significant inefficiency.

2. **Lack of structural interpretability in intermediate states.** As shown in Fig. 5a, intermediate solutions during denoising are usually dense and noisy (e.g., full graphs for TSP), offering little insight into how variables are progressively determined.

The recent introduction of **consistency models** [25] has mitigated the first issue by enabling high-quality solutions with as few as 3-5 inference steps, refered as FastT2T [24]. Building on this progress, our work addresses the second issue by proposing a **structured diffusion framework** with step-wise interpretability.

As shown in Fig. 5b, our model begins from an empty solution and progressively activates decision variables with high predicted confidence at each inference step. This process can be viewed as a variable selection trajectory, where each intermediate state incrementally extends the previous one, forming a structurally consistent sequence of partial solutions.

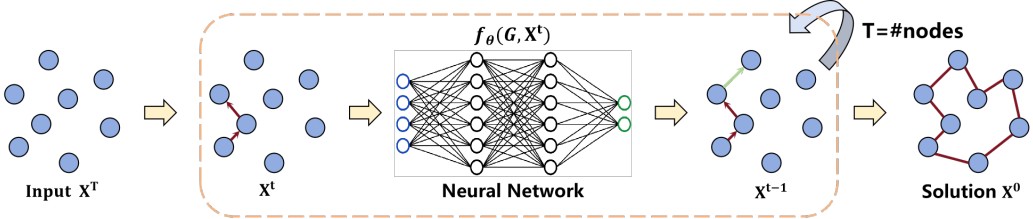

(a) The solving pipeline of Autoregressive construction.

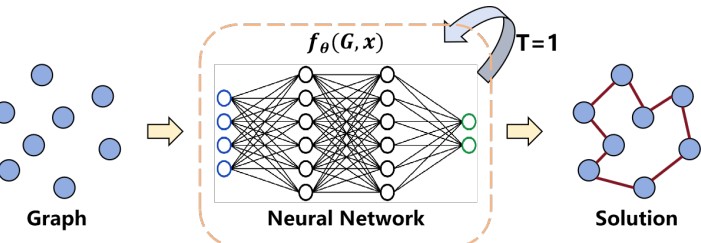

(b) The solving pipeline of Non-Autoregressive construction.

Figure 4: Comparison of solving pipelines between Autoregressive and Non-Autoregressive constructions.

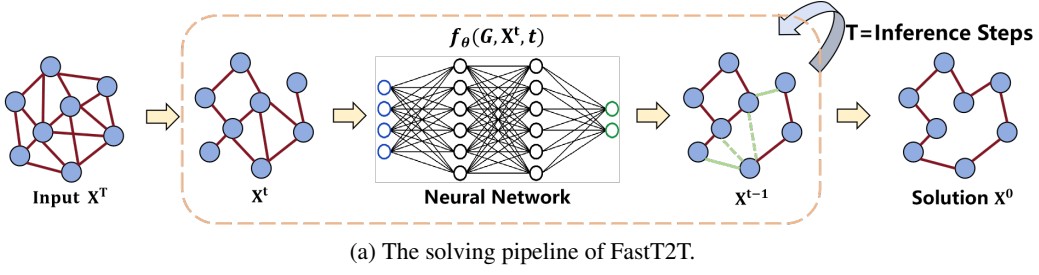

(a) The solving pipeline of FastT2T.

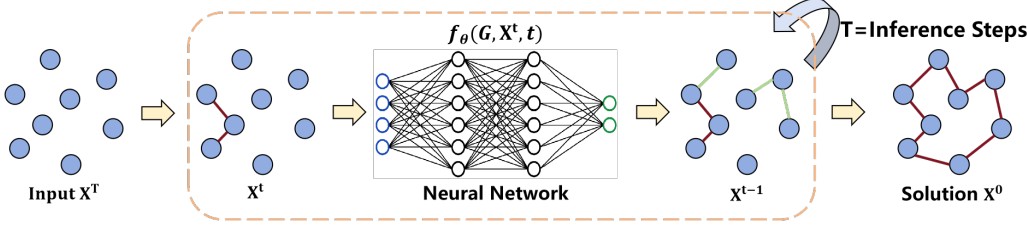

(b) The solving pipeline of StruDiCO (Ours).

Figure 5: Comparison of solving pipelines between FastT2T and StruDiCO. Green solid lines indicate newly added edges, while green dashed lines represent deleted edges.

# B   Discussion on Structural Consistency in Denoising

## B.1   Limitations of Vanilla Reverse Sampling

Most diffusion-based CO solvers, including our main pipeline (see Sec. 3.2), follow the standard multistep consistency sampling protocol [24, 25], as shown in Algorithm 3, which iteratively denoises and resamples $\mathbf{x}_0$ given intermediate states $\mathbf{x}_\tau$. While this process can recover high-quality solutions, it does not strictly enforce any structural constraints on the reverse trajectory, such as **feasibility preservation** across steps.

---

**Algorithm 3** Multistep Consistency Sampling

---

**Require:** Consistency model $f_\theta(\cdot, \cdot, \cdot)$, graph instance $G$, time sequence $\tau_1 > \cdots > \tau_{N_\tau - 1}$
 1: Sample $\mathbf{x}_T \sim \mathcal{U}$
 2: $p_\theta(\mathbf{x}_0 \mid G) \leftarrow f_\theta(\mathbf{x}_T, T, G)$
 3: $\mathbf{x}_0 \sim p_\theta(\mathbf{x}_0 \mid G)$
 4: **for** $n = 1$ to $N_\tau - 1$ **do**
 5:     $\mathbf{x}_{\tau_n} \sim \text{Cat}(\mathbf{p} = \tilde{\mathbf{x}}_0 \overline{\mathbf{Q}}_{\tau_n})$
 6:     $p_\theta(\mathbf{x}_0 \mid G) \leftarrow f_\theta(\mathbf{x}_{\tau_n}, \tau_n, G)$
 7:     $\mathbf{x}_0 \sim p_\theta(\mathbf{x}_0 \mid G)$
 8: **end for**
 9: **return** Solution $\mathbf{x}_0$

---

## B.2   Step-Wise Decoding Sampling for Intermediate Feasibility

To further improve step-wise feasibility, we explore a stronger variant: **step-wise greedy decoding**. At each reverse step $\tau$, we rank the predicted heatmap $p_\theta(\mathbf{x}_0 = 1 \mid \mathbf{x}_\tau, \tau, G)$ in descending order and activate the top $n_\tau$ variables to form $\mathbf{x}_\tau$, subject to task-specific constraints (e.g., no subtours or conflicts). Here, $n_\tau$ is determined by the expected number of active variables given by the cumulative transition matrix $\overline{\mathbf{Q}}_\tau$ under the reverse schedule.

This approach better aligns the reverse process with the semantics of the variable-absorption forward process and can be combined with feasibility filters (e.g., constraint checks) to construct strictly valid $\mathbf{x}_\tau$ at every step. However, systematic exploration of this strategy is left as future work.

# C  Full Proof of Theorem 3.1

We provide the complete mathematical proof of Theorem 3.1, which formally analyzes the effect of threshold-based masking on the KL divergence between the predicted and true solution distributions.

**Theorem C.1** (Constrained Sampling Reduces Divergence, Full Version). *Let $q(\mathbf{x}_0 \mid G)$ be the true distribution and $p_\theta(\mathbf{x}_0 \mid \mathbf{x}_\tau, \tau, G)$ the model prediction. Fix a threshold $\delta \in (0, 1)$ and define the constrained support*

$$\mathcal{M}_\delta = \{i \in [N] : p_\theta(\mathbf{x}_0^i = 1 \mid \mathbf{x}_\tau, \tau, G) > \delta\}, \tag{7}$$

$$\mathcal{S}_\delta = \{\mathbf{x}_0 : \operatorname{supp}(\mathbf{x}_0) \subseteq \mathcal{M}_\delta\}. \tag{8}$$

*Let the masked (renormalized) model be*

$$\tilde{p}_\theta(\mathbf{x}_0) = \frac{p_\theta(\mathbf{x}_0) \, \mathbb{I}[\mathbf{x}_0 \in \mathcal{S}_\delta]}{Z_\delta}, \tag{9}$$

$$Z_\delta = \sum_{\mathbf{x}_0 \in \mathcal{S}_\delta} p_\theta(\mathbf{x}_0) \in (0, 1], \tag{10}$$

*and define the log-likelihood ratio $\ell(\mathbf{x}_0) = \log \frac{p_\theta(\mathbf{x}_0)}{q(\mathbf{x}_0)}$. Assume $q(\mathbf{x}_0) > 0$ whenever $p_\theta(\mathbf{x}_0) > 0$ (so that $D_{\mathrm{KL}}(p_\theta \| q) < \infty$). Then the reverse-KL difference admits the exact decomposition*

$$D_{\mathrm{KL}}(\tilde{p}_\theta \| q) = D_{\mathrm{KL}}(p_\theta \| q) - (1 - Z_\delta)\big(\Delta_\delta + c_\delta\big), \tag{11}$$

*where*

$$\Delta_\delta = \mathbb{E}_{p_\theta(\cdot \mid \mathcal{S}_\delta^c)}[\ell(\mathbf{x}_0)] - \mathbb{E}_{p_\theta(\cdot \mid \mathcal{S}_\delta)}[\ell(\mathbf{x}_0)], \tag{12}$$

$$c_\delta = \frac{-\log Z_\delta}{1 - Z_\delta} \quad (\geq 0). \tag{13}$$

*Consequently,*

$$D_{\mathrm{KL}}(\tilde{p}_\theta \| q) \leq D_{\mathrm{KL}}(p_\theta \| q) + \varepsilon_\delta, \tag{14}$$

$$\varepsilon_\delta = (1 - Z_\delta)\big[-(\Delta_\delta + c_\delta)\big]_+, \tag{15}$$

*and in particular, if $\Delta_\delta \geq -c_\delta$ then $D_{\mathrm{KL}}(\tilde{p}_\theta \| q) \leq D_{\mathrm{KL}}(p_\theta \| q)$, with strict inequality whenever $Z_\delta < 1$ and the inequality is strict.*

*Proof.* All sums below are over the discrete space of $\mathbf{x}_0$. We use the shorthand $p(\mathbf{x}_0) = p_\theta(\mathbf{x}_0 \mid \mathbf{x}_\tau, \tau, G)$, $q(\mathbf{x}_0) = q(\mathbf{x}_0 \mid G)$, and $\tilde{p}(\mathbf{x}_0) = \tilde{p}_\theta(\mathbf{x}_0)$, suppressing the conditioning for readability.

**Step 1: Reverse-KL under masking.** By definition of $\tilde{p}$ and $Z_\delta$,

$$\tilde{p}(\mathbf{x}_0) = \begin{cases} \dfrac{p(\mathbf{x}_0)}{Z_\delta}, & \mathbf{x}_0 \in \mathcal{S}_\delta, \\ 0, & \mathbf{x}_0 \notin \mathcal{S}_\delta. \end{cases} \tag{16}$$

Since $q > 0$ wherever $p > 0$, we have $q > 0$ on $\mathcal{S}_\delta$ (because $p > 0$ there if and only if $\tilde{p} > 0$). Then

$$D_{\mathrm{KL}}(\tilde{p} \| q) = \sum_{\mathbf{x}_0 \in \mathcal{S}_\delta} \tilde{p}(\mathbf{x}_0) \log \frac{\tilde{p}(\mathbf{x}_0)}{q(\mathbf{x}_0)} \tag{17}$$

$$= \sum_{\mathbf{x}_0 \in \mathcal{S}_\delta} \frac{p(\mathbf{x}_0)}{Z_\delta} \log \frac{p(\mathbf{x}_0)/Z_\delta}{q(\mathbf{x}_0)} \tag{18}$$

$$= \sum_{\mathbf{x}_0 \in \mathcal{S}_\delta} \frac{p(\mathbf{x}_0)}{Z_\delta} \log \frac{p(\mathbf{x}_0)}{q(\mathbf{x}_0)} - \log Z_\delta \sum_{\mathbf{x}_0 \in \mathcal{S}_\delta} \frac{p(\mathbf{x}_0)}{Z_\delta} \tag{19}$$

$$= \mathbb{E}_{p(\cdot \mid \mathcal{S}_\delta)}\left[\log \frac{p(\mathbf{x}_0)}{q(\mathbf{x}_0)}\right] - \log Z_\delta \tag{20}$$

$$= \mathbb{E}_{p(\cdot \mid \mathcal{S}_\delta)}[\ell(\mathbf{x}_0)] - \log Z_\delta. \tag{21}$$

**Step 2: Reverse-KL of the original model.** Decompose $p$ into its mass on $\mathcal{S}_\delta$ and $\mathcal{S}_\delta^c$:

$$D_{\mathrm{KL}}(p\|q) = \sum_{\mathbf{x}_0} p(\mathbf{x}_0)\,\log\frac{p(\mathbf{x}_0)}{q(\mathbf{x}_0)} \tag{22}$$

$$= \sum_{\mathbf{x}_0\in\mathcal{S}_\delta} p(\mathbf{x}_0)\,\ell(\mathbf{x}_0) \;+\; \sum_{\mathbf{x}_0\notin\mathcal{S}_\delta} p(\mathbf{x}_0)\,\ell(\mathbf{x}_0) \tag{23}$$

$$= Z_\delta\,\mathbb{E}_{p(\cdot|\mathcal{S}_\delta)}[\ell(\mathbf{x}_0)] \;+\; (1-Z_\delta)\,\mathbb{E}_{p(\cdot|\mathcal{S}_\delta^c)}[\ell(\mathbf{x}_0)]. \tag{24}$$

**Step 3: Taking the difference.** Subtracting (B) from (A) gives

$$D_{\mathrm{KL}}(\tilde{p}\|q) - D_{\mathrm{KL}}(p\|q) \tag{25}$$

$$= \Big(\mathbb{E}_{p(\cdot|\mathcal{S}_\delta)}[\ell] - \log Z_\delta\Big) - \Big(Z_\delta\,\mathbb{E}_{p(\cdot|\mathcal{S}_\delta)}[\ell] + (1-Z_\delta)\,\mathbb{E}_{p(\cdot|\mathcal{S}_\delta^c)}[\ell]\Big) \tag{26}$$

$$= (1-Z_\delta)\,\mathbb{E}_{p(\cdot|\mathcal{S}_\delta)}[\ell] - (1-Z_\delta)\,\mathbb{E}_{p(\cdot|\mathcal{S}_\delta^c)}[\ell] - \log Z_\delta \tag{27}$$

$$= -(1-Z_\delta)\Big(\underbrace{\mathbb{E}_{p(\cdot|\mathcal{S}_\delta^c)}[\ell] - \mathbb{E}_{p(\cdot|\mathcal{S}_\delta)}[\ell]}_{\Delta_\delta}\Big) - \log Z_\delta \tag{28}$$

$$= -(1-Z_\delta)\,\Delta_\delta \;-\; \log Z_\delta. \tag{29}$$

When $Z_\delta = 1$ the mask is identity and both sides are equal; for $Z_\delta \in (0,1)$ rewrite the last term as

$$-\log Z_\delta = (1-Z_\delta)\,\frac{-\log Z_\delta}{1-Z_\delta} = (1-Z_\delta)\,c_\delta, \tag{30}$$

which yields the exact decomposition Eq. (11):

$$D_{\mathrm{KL}}(\tilde{p}\|q) = D_{\mathrm{KL}}(p\|q) - (1-Z_\delta)\big(\Delta_\delta + c_\delta\big). \tag{31}$$

**Step 4: Upper bound and sufficient condition.** Since for any real $x$, $x = x_+ - (-x)_+$ with $x_+ = \max\{x,0\}$, we get the one-sided bound

$$D_{\mathrm{KL}}(\tilde{p}\|q) \le D_{\mathrm{KL}}(p\|q) + (1-Z_\delta)\big[-(\Delta_\delta + c_\delta)\big]_+. \tag{32}$$

Moreover, whenever $\Delta_\delta \ge -c_\delta$ (equivalently $\mathbb{E}_{p(\cdot|\mathcal{S}_\delta^c)}[\ell] \ge \mathbb{E}_{p(\cdot|\mathcal{S}_\delta)}[\ell] - c_\delta$), the difference in Eq. (11) is nonpositive and thus $D_{\mathrm{KL}}(\tilde{p}\|q) \le D_{\mathrm{KL}}(p\|q)$, with strict improvement if $Z_\delta < 1$ and the inequality is strict. $\qquad\square$

# D  Supplementary Experiments

## D.1  Applied to Capacitated Vehicle Routing Problem

To further demonstrate the generality of StruDiCO beyond classical edge-selection (TSP) and node-selection (MIS) tasks, we extend it to the Capacitated Vehicle Routing Problem (CVRP), a more challenging CO task that involves both global routing structures and hard feasibility constraints, such as customer demands and vehicle capacities. The refinement procedure for CVRP follows the same design as that for TSP: greedy decoding is performed based on the learned heatmap, with the objective guided by a distance-based penalty term. We evaluate StruDiCO on CVRP-50, CVRP-100, and CVRP-200 under standard benchmark settings [16]. To further improve both solution feasibility and cost, we incorporate a local search heuristic (Classic-LS [45]) during inference. As summarized in Table 7, these results demonstrate that StruDiCO effectively generalizes to CVRP, consistently improving over learning-based baselines while remaining efficient.

## D.2  Results on Real-World Data

We evaluate our model trained on randomly generated 100-node instances against real-world TSPLIB benchmarks containing 50-200 nodes. For fair comparison, we adopt the same hyperparameter settings as [24], with $T_s = T_g = 10$. As shown in Table 8, StruDiCO demonstrates superior performance, achieving an average optimality gap of only 0.13%.

Table 7: Results on CVRP. LS: Local Search.

| Algorithm | Type | CVRP 50 | | | CVRP 100 | | | CVRP 200 | | |
|---|---|---|---|---|---|---|---|---|---|---|
| | | Length↓ | Drop↓ | Time | Length↓ | Drop↓ | Time | Length↓ | Drop↓ | Time |
| HGS [46] | Heuristics | 10.37 | 0.00% | 1s | 15.56 | 0.00% | 20s | 19.63 | 0.00% | 60s |
| Sym-NCO [16] | RL+LS | 10.57 | 1.95% | 0.09s | 15.93 | 2.37% | 0.19s | 20.19 | 2.86% | 0.36s |
| COExpander [47] | SL+LS | 10.77 | 3.90% | 0.04s | 16.22 | 4.25% | 0.06s | 20.59 | 4.89% | 0.15s |
| StruDiCO ($T_s$=3,$T_g$=3) | SL+LS | 10.48 | 1.12% | 0.05s | 15.85 | 1.88% | 0.11s | 20.25 | 3.19% | 0.26s |
| StruDiCO ($T_s$=5,$T_g$=5) | SL+LS | 10.45 | 0.85% | 0.07s | 15.80 | 1.53% | 0.17s | 20.16 | 2.71% | 0.38s |

Table 8: Solution quality for methods trained on random 100-node problems and evaluated on **TSPLIB instances with 50-200 nodes**. * denotes results quoted from previous works [24].

| Instances | AM* | GCN* | Learn2OPT* | GNNGLS* | DIFUSCO* | T2T* | Fast T2T* | Ours |
|---|---|---|---|---|---|---|---|---|
| eil51 | 16.767% | 40.025% | 1.725% | 1.529% | 2.82% | 0.14% | 0.00% | 0.00% |
| berlin52 | 4.169% | 33.225% | 0.449% | 0.142% | 0.00% | 0.00% | 0.00% | 0.00% |
| st70 | 1.737% | 24.785% | 0.040% | 0.764% | 0.00% | 0.00% | 0.00% | 0.00% |
| eil76 | 1.992% | 27.411% | 0.096% | 0.163% | 0.34% | 0.00% | 0.00% | 0.00% |
| pr76 | 0.816% | 27.793% | 1.228% | 0.039% | 1.12% | 0.40% | 0.00% | -0.00% |
| rat99 | 2.645% | 17.633% | 0.123% | 0.550% | 0.09% | 0.09% | 0.00% | 0.00% |
| kroA100 | 4.017% | 28.828% | 18.313% | 0.728% | 0.10% | 0.00% | 0.00% | 0.00% |
| kroB100 | 5.142% | 34.686% | 1.119% | 0.147% | 2.29% | 0.74% | 0.65% | 0.00% |
| kroC100 | 0.972% | 35.506% | 0.349% | 1.571% | 0.00% | 0.00% | 0.00% | 0.00% |
| kroD100 | 2.717% | 38.018% | 0.866% | 0.572% | 0.07% | 0.00% | 0.00% | 0.00% |
| kroE100 | 1.470% | 26.589% | 1.832% | 1.216% | 3.83% | 0.27% | 0.13% | 2.15% |
| rd100 | 3.407% | 50.432% | 1.725% | 0.003% | 0.08% | 0.00% | 0.00% | 0.11% |
| eil101 | 2.994% | 21.776% | 1.529% | 0.03% | 0.03% | 0.00% | 0.00% | 0.00% |
| lin105 | 1.739% | 34.902% | 1.867% | 0.606% | 0.00% | 0.00% | 0.00% | 0.18% |
| pr107 | 3.933% | 80.564% | 0.898% | 0.439% | 0.91% | 0.61% | 0.62% | 0.18% |
| pr124 | 2.677% | 70.146% | 10.232% | 0.755% | 1.02% | 0.08% | 0.08% | -0.00% |
| bier127 | 5.908% | 45.561% | 3.044% | 1.948% | 0.94% | 0.54% | 1.50% | 0.04% |
| ch130 | 3.182% | 39.090% | 0.709% | 3.519% | 0.29% | 0.06% | 0.00% | 0.24% |
| pr136 | 5.064% | 58.673% | 0.000% | 3.387% | 0.19% | 0.10% | 0.01% | 0.04% |
| pr144 | 7.641% | 55.837% | 1.526% | 3.581% | 0.80% | 0.50% | 0.39% | 0.00% |
| ch150 | 1.584% | 49.743% | 0.321% | 2.113% | 0.57% | 0.49% | 0.00% | 0.04% |
| kroA150 | 3.784% | 45.411% | 0.724% | 2.984% | 0.34% | 0.14% | 0.00% | 0.24% |
| kroB150 | 2.437% | 56.745% | 0.886% | 3.258% | 0.30% | 0.00% | 0.07% | 0.00% |
| pr152 | 7.494% | 49.376% | 3.119% | 3.119% | 1.69% | 0.83% | 0.19% | 0.69% |
| u159 | 7.551% | 38.338% | 0.054% | 1.020% | 0.82% | 0.00% | 0.00% | 0.00% |
| rat195 | 6.893% | 24.968% | 0.743% | 1.666% | 1.48% | 1.27% | 0.79% | -0.00% |
| d198 | 373.020% | 62.952% | 0.522% | 4.727% | 3.32% | 1.97% | 0.86% | -0.00% |
| kroA200 | 7.106% | 40.885% | 1.441% | 2.029% | 2.28% | 0.57% | 0.49% | 0.00% |
| kroB200 | 8.541% | 43.643% | 0.646% | 2.589% | 2.35% | 0.92% | 2.50% | -0.00% |
| **Mean** | **16.767%** | **40.025%** | **1.725%** | **1.529%** | **0.97%** | **0.35%** | **0.28%** | **0.13%** |

### D.3 Supplementary Results of Generalization Study

To further evaluate the generalization ability of StruDiCO, we conduct additional experiments on MIS instances with varying edge densities and graph sizes. Specifically, we test models trained on ER graphs with $p = 0.15$ and $n = 700$–$800$ on out-of-distribution graphs with different $p$ values (0.2, 0.3, 0.4) and node counts (350–400, 1400–1600). As shown in Figures 6–9, StruDiCO consistently outperforms prior diffusion-based baselines (e.g., DIFUSCO [22], T2T [23], Fast T2T [24]) under both greedy and sampling decoding, demonstrating superior adaptability to distribution shifts in both graph sparsity and scale.

## E   Experimetnal Details

### E.1   Hardware

All models are trained and tested using NVIDIA A40 (48G) GPUs and Intel(R) Xeon(R) Gold 5220 CPU @ 2.20GHz.

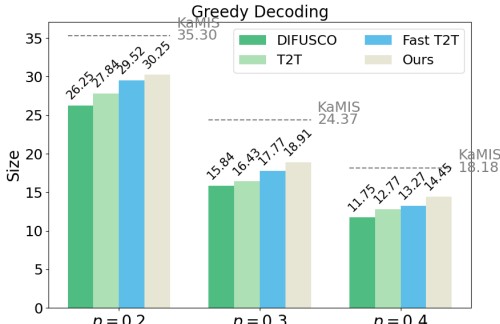

Figure 6: Generalization performance from $p = 0.15$ to $p = 0.2$, $p = 0.3$, and $p = 0.4$.

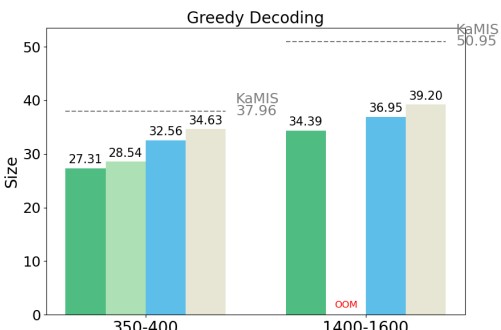

Figure 7: Generalization performance from ER 700–800 to ER 350–400 and 1400–1600.

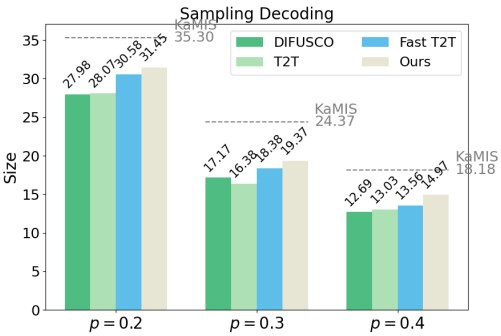

Figure 8: Generalization performance from $p = 0.15$ to $p = 0.2$, $p = 0.3$, and $p = 0.4$.

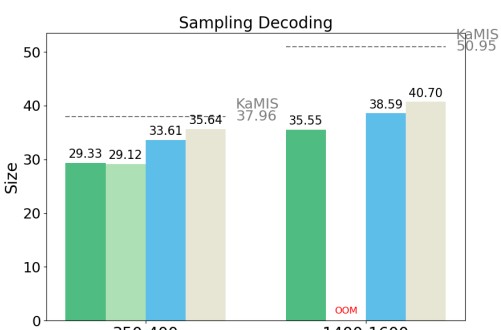

Figure 9: Generalization performance from ER 700–800 to ER 350–400 and 1400–1600.

## E.2  Sparsification

For large-scale TSP problems, we follow [22, 23, 24] to employ sparse graphs, as sparsified by constraining each node to connect to only its k nearest neighbors, determined by Euclidean distances. For TSP-500, we set $k = 50$, and for TSP-1000, $k = 100$. This strategy prevents the exponential increase in edges typical in dense graphs as node count rises.

## E.3  Dataset

Our dataset settings follow previous works [22, 23, 24] to ensure fair and consistent comparisons. The reference solutions for TSP-50/100 are labeled using the Concorde exact solver [36], while TSP-500/1000 solutions are generated by the LKH-3 heuristic solver [26]. The test sets for TSP-50/100 contain 1280 instances sourced from [12, 17], and those for TSP-500/1000 contain 128 instances from [18]. For the MIS task, we use KaMIS [38] to generate reference solutions for both RB and ER graphs. The RB training set consists of 90,000 randomly generated instances, and the test set includes 500 instances. For ER graphs, we generate 163,840 training instances, and the test set is obtained from [19].

## E.4  Hyperparameters

We have organized the training settings and model parameters of StruDiCO in Table 9.

Table 9: Details about the training hyperparameters of StruDiCO.

| Problem | Dataset Size | Batch Size | Epoch | Learning Rate | Hidden Dimension | Noise Degree |
|---------|-------------|-----------|-------|---------------|------------------|--------------|
| TSP-50 | 1,502k | 32 | 50 | 2e-4 | 256 | 0.5 |
| TSP-100 | 1,502k | 12 | 50 | 2e-4 | 256 | 0.5 |
| TSP-500 | 128k | 6 | 50 | 2e-4 | 256 | 0.5 |
| TSP-1000 | 64k | 4 | 50 | 2e-4 | 256 | 0.5 |
| MIS-RB | 90k | 4 | 50 | 2e-4 | 256 | 0.3 |
| MIS-ER | 163k | 4 | 50 | 2e-4 | 128 | 0.5 |

# F Network Architecture Details

## F.1 Input Embedding Layer

Given node vector $x \in \mathbb{R}^{N \times 2}$, weighted edge vector $e \in \mathbb{R}^E$, denoising timestep $t \in \{\tau_1, \ldots, \tau_M\}$, where $N$ denotes the number of nodes in the graph, and $E$ denotes the number of edges, we compute the sinusoidal features of each input element respectively:

$$\tilde{x}_i = \text{concat}(\tilde{x}_{i,0}, \tilde{x}_{i,1}) \tag{21}$$

$$\tilde{x}_{i,j} = \text{concat}\left(\sin \frac{x_{i,j}}{T^{0/d}}, \cos \frac{x_{i,j}}{T^{0/d}}, \sin \frac{x_{i,j}}{T^{2/d}}, \cos \frac{x_{i,j}}{T^{2/d}}, \ldots, \sin \frac{x_{i,j}}{T^{d/d}}, \cos \frac{x_{i,j}}{T^{d/d}}\right) \tag{22}$$

$$\tilde{e}_i = \text{concat}\left(\sin \frac{e_i}{T^{0/d}}, \cos \frac{e_i}{T^{0/d}}, \sin \frac{e_i}{T^{2/d}}, \cos \frac{e_i}{T^{2/d}}, \ldots, \sin \frac{e_i}{T^{d/d}}, \cos \frac{e_i}{T^{d/d}}\right) \tag{23}$$

$$\tilde{t} = \text{concat}\left(\sin \frac{t}{T^{0/d}}, \cos \frac{t}{T^{0/d}}, \sin \frac{t}{T^{2/d}}, \cos \frac{t}{T^{2/d}}, \ldots, \sin \frac{t}{T^{d/d}}, \cos \frac{t}{T^{d/d}}\right) \tag{24}$$

where $d$ is the embedding dimension, $T$ is a large number (usually selected as 10000), and concat($\cdot$) denotes concatenation.

Next, we compute the input features of the graph convolution layer:

$$x_i^0 = W_1^0 \tilde{x}_i \tag{25}$$

$$e_i^0 = W_2^0 \tilde{e}_i \tag{26}$$

$$t^0 = W_4^0(\text{ReLU}(W_3^0 \tilde{t})) \tag{27}$$

where $t^0 \in \mathbb{R}^{d_t}$, $d_t$ is the time feature embedding dimension. Specifically, for TSP, the embedding input edge vector $e$ is a weighted adjacency matrix, which represents the distance between different nodes, and $e^0$ is computed as above. For MIS, we initialize $e^0$ to a zero matrix $0^{E \times d}$.

## F.2 Graph Convolution Layer

Following [17], the cross-layer convolution operation is formulated as:

$$x_i^{l+1} = x_i^l + \text{ReLU}(\text{BN}(W_1^l x_i^l + \sum_{j \sim i} \eta_{ij}^l \odot W_2^l x_j^l)) \tag{28}$$

$$e_{ij}^{l+1} = e_{ij}^l + \text{ReLU}(\text{BN}(W_3^l e_{ij}^l + W_4^l x_i^l + W_5^l x_j^l)) \tag{29}$$

$$\eta_{ij}^l = \frac{\sigma(e_{ij}^l)}{\sum_{j' \sim i} \sigma(e_{ij'}^l) + \epsilon} \tag{30}$$

where $x_i^l$ and $e_{ij}^l$ denote the node feature vector and edge feature vector at layer $l$, $W_1, \cdots, W_5 \in \mathbb{R}^{h \times h}$ denote the model weights, and $\eta_{ij}^l$ denotes the dense attention map. The convolution operation integrates the edge feature to accommodate the significance of edges in routing problems.

For TSP, we aggregate the timestep feature with the edge convolutional feature and reformulate the update for edge features as follows:

$$e_{ij}^{l+1} = e_{ij}^l + \text{ReLU}(\text{BN}(W_3^l e_{ij}^l + W_4^l x_i^l + W_5^l x_j^l)) + W_6^l(\text{ReLU}(t^0)) \tag{31}$$

For MIS, we aggregate the timestep feature with the node convolutional feature and reformulate the update for node features as follows:

$$x_i^{l+1} = x_i^l + \text{ReLU}(\text{BN}(W_1^l x_i^l + \sum_{j \sim i} \eta_{ij}^l \odot W_2^l x_j^l)) + W_6^l(\text{ReLU}(t^0)) \tag{32}$$

### F.3  Output Layer

The prediction of the edge heatmap in TSP and node heatmap in MIS is as follows:

$$e_{i,j} = \text{Softmax}(\text{norm}(\text{ReLU}(W_e e_{i,j}^L))) \tag{33}$$
$$x_i = \text{Softmax}(\text{norm}(\text{ReLU}(W_n x_i^L))) \tag{34}$$

where $L$ is the number of GCN layers and norm is layer normalization.

### F.4  Hyper-parameters

For both TSP and MIS tasks, we construct a 12-layer GCN derived above. We set the node, edge, and timestep embedding dimension $d = 256, 128$ for TSP and MIS tasks, respectively.

## G  Limitations and Broader Impacts

Despite the effectiveness of our proposed method, several limitations remain. First, our model is trained under a supervised learning paradigm, which relies on access to a substantial number of labeled optimal or near-optimal solutions. While this setting is practical for many standard CO benchmarks, it may present limitations when extending to domains where high-quality supervision is difficult or expensive to obtain. Second, although we demonstrate improved generalization over existing diffusion-based solvers [24] in fixed-scale settings, the model still faces challenges in handling real-world CO problems with unknown or variable instance sizes. Third, while our diffusion process incrementally constructs solutions in a structured manner, it still follows a non-autoregressive paradigm and thus requires heuristic post-processing (e.g., greedy decoding) to enforce final feasibility. This dependency may constrain its applicability in tasks where domain-specific heuristics are unavailable or expensive to design.

This work contributes to the growing field of learning-based combinatorial optimization, with potential applications in logistics, circuit design, scheduling, and operations research. By introducing interpretable and efficient generative methods, our approach can improve decision-making in domains that require both scalability and transparency.

