# OpenReview forum: "StruDiCO: Structured Denoising Diffusion with Gradient-free Inference-stage Boosting for Memory and Time Efficient Combinatorial Optimization"
_NeurIPS.cc/2025/Conference — NeurIPS 2025 poster_

### Official Review · Reviewer_2ZPS · 2025-06-23

**Clarity:** 3
**Significance:** 3
**Originality:** 4
**Rating:** 5
**Confidence:** 4

**Summary:**

This work focuses on designing diffusion-based solvers for combinatorial optimization (CO) problems. Compared to recent diffusion solvers, its main innovation lies in a novel structure-erasing noising strategy: instead of adding discrete noise directly to label data during the forward process, the model randomly masks a subset of edges in the solution graph. During the reverse denoising process, the model starts from an empty graph and progressively predicts which edges should be recovered to reconstruct a complete solution.

Building on this structured modeling approach, the authors further introduce a gradient-free post-optimization strategy, which repeatedly masks parts of the current solution and refines it through a guided search process using a custom-designed scoring function (confidence divided by task objective). The authors claim their method achieves state-of-the-art (SOTA) performance in both solution quality and inference speed.

**Questions:**

Some recent diffusion-based solvers have been overlooked in the comparison, such as DISCO: Efficient Diffusion Solver for Large-Scale Combinatorial Optimization Problems, which also explores improving both solution quality and inference speed for CO problems. A technical discussion of the differences from these works, along with public performance comparisons, would further strengthen the completeness and positioning of this work.

**Ethical Concerns:**

["NO or VERY MINOR ethics concerns only"]

**Final Justification:**

The reviewer appreciates the detailed and thoughtful response. All of my concerns have been addressed. In particular, I am impressed by StruDiCO’s strong performance on large-scale combinatorial optimization problems, which clearly demonstrates the method’s scalability. I believe this work establishes a new state-of-the-art in the CO domain. Based on these clarifications and results, I am updating my score from 4 to 5.

**Limitations:**

The reviewer finds the idea of a structured denoising process compelling and is generally supportive of the paper. Ablation studies should be included to clarify the contributions of key components. The current score may be raised depending on the authors’ responses in the rebuttal phase.

**Paper Formatting Concerns:**

The reviewer has no concerns regarding the formatting of the paper.

**Quality:**

3

**Strengths And Weaknesses:**

Strengths:
1. The reviewer appreciates the idea of structurally designing the diffusion process. While diffusion models have recently shown strong potential as general-purpose function approximators in CO, prior diffusion solvers typically add discrete noise directly to the initial solution, which leads to a lack of interpretability in both the noising and denoising stages, limiting the integration of search-based enhancements. The structured design proposed here alleviates this issue.
2. The proposed solver outperforms recent neural CO solvers in both solution quality and inference speed, validating the effectiveness of the structured solver design tailored for combinatorial optimization.
3. Compared to gradient-based search strategies, the gradient-free post-optimization approach introduced here is more convenient to implement and apply, highlighting StruDiCO's practical advantage in broader applications.

Weaknesses:
1. The reviewer is curious about the contribution of the two key diffusion design components, Variable-Absorption and Constrained Consistency Sampling, to the overall performance. Conducting a thorough ablation study would help clarify their individual effects.
2. Whether the inference cost of StruDiCO increases significantly with problem size, such as TSP-10000. While this does not affect the rating, clarifying this point would be helpful.

---

> ### Author Rebuttal · Authors · 2025-07-26
>
> Dear Reviewer 2ZPS,
>
> Thanks for your insightful comments and for acknowledging novelty, model design, and empirical performance. Below we respond to the specific comments.
>
> ---
>
> > **Q1&W2: Some recent diffusion-based solvers have been overlooked in the comparison, such as DISCO: Efficient Diffusion Solver for Large-Scale Combinatorial Optimization Problems, which also explores improving both solution quality and inference speed for CO problems. A technical discussion of the differences from these works, along with public performance comparisons, would further strengthen the completeness and positioning of this work. Whether the inference cost of StruDiCO increases significantly with problem size, such as TSP-10000. While this does not affect the rating, clarifying this point would be helpful.**
>
> Thanks for providing the related work. DISCO improves inference efficiency and solution quality on large-scale CO problems (e.g., TSP, MIS) via residue-constrained denoising and multi-modal graph search. We have now incorporated a discussion of DISCO in the related work section.
>
> We have extended our model training to **TSP-10,000** and compared it with diffusion-based baselines, including **DIFUSCO** and **DISCO**, and a  divide-and-conquer (DC) approach, **GLOP** [1].
>
>
> | Method                         | Type         | Length ↓ | Drop ↓  | Time  |
> |--------------------------------|--------------|----------|---------|--------|
> | LKH           | Heuristics   | 71.77    | 0.00%   | 332s   |
> | GLOP (more revisions)          | DC           | 75.29    | 4.90%   | 15s    |
> | DIFUSCO ($T_s = 100$)             | SL+G+2OPT    | 73.91    | 2.98%   | 124s   |
> | DISCO                          | SL+G+2OPT    | 73.84    | 2.88%   | 92s    |
> | **StruDiCO ($T_s = 5$)**          | SL+G+2OPT    | **72.90**| **1.57%** | **60s** |
>
> It is important to note that as the problem size increases (e.g., 100K nodes), the primary bottleneck during inference is not the diffusion (or consistency) model itself. Rather, **the major computational overhead arises from the greedy decoding procedure**, where edges or nodes with the highest confidence are sequentially inserted into a partial solution until a feasible one is constructed. This step can become prohibitively time-consuming in large-scale instances. We believe that the scalability of StruDiCO could be further improved by combining its backbone generative model with a more efficient decoding strategy or a  divide-and-conquer approach. This presents a promising direction for future research.
>
> We will include these large-scale results and comparisons with DISCO in the final version to strengthen the contribution.
>
> ---
>
> > **W1: The reviewer is curious about the contribution of the two key diffusion design components, Variable-Absorption and Constrained Consistency Sampling, to the overall performance. Conducting a thorough ablation study would help clarify their individual effects.**
>
> We provide a detailed ablation study to isolate the contribution of each component in our method. The analysis focuses on three parts:
>
> 1. The role of **Variable-Absorption (VA)** mechanism,
> 2. The effect of **Constrained Consistency Sampling (CCS)** in the main inference process, and
> 3. The contribution of the **Objective-Aware Refinement** module applied on the initial solution solved by StruDiCO with $T_s=5$.
>
> Specifically, we compare several variants, including:
> - **Uniform CM**: the uniform cosistency model (CM) proposed in Fast T2T;
> - **VA CM**: replacing the uniform CM with the Variable-Absorption CM;
> - **+CCS**: where low-confidence variables are masked;
> - **Vanilla Refinement**: which refines the initial solution without gradient-free objective guidance (OG).
>
> #### Table: Ablation Study (Gap, % ↓)
>
> | Variant                      | TSP-50 | TSP-100 | TSP-500 | TSP-1000 | MIS-RB | MIS-ER |
> |-----------------------------|--------|---------|---------|----------|--------|--------|
> | Uniform CM (Fast T2T, \$T\_s\$=5)       | 0.031     | 0.101     | 0.922     | 1.167     | 6.023     | 15.898     |
> | VA CM (StruDiCO)               | 0.032  | 0.163   | 1.091   | 1.183    | 7.476  | 23.461 |
> |Uniform +CCS |     0.025 | 0.081   | 0.898   | 1.074  | 5.878  | 16.522 |
> | VA + CCS (StruDiCO, \$T\_s\$=5)      | 0.019  | 0.067   | 0.728   | 0.797    | 3.899  | 11.450 |
> | Vanilla Refinement          | 0.012  | 0.033   | 0.381   | 0.672    | 1.922  | 7.611  |
> | + OG (StruDiCO, \$T\_s\$=5, \$T\_g\$=5)    | **0.011** | **0.025** | **0.319** | **0.473**  | **1.707** | **5.579** |
>
> As shown, replacing the uniform noise (used in DIFUSCO, T2T, and FastT2T) with the Variable-Absorption (VA) mechanism alone leads to a performance degradation. This is because VA introduces a monotonic absorption process, where variables are only deactivated (i.e., 1 → 0) during the forward diffusion. While this preserves structural continuity at each timestep, it results in overly conservative trajectories when used alone—**limiting exploration and diversity in the reverse process**. In contrast, the Uniform Consistency Model (CM) allows symmetric noise injection (both 0 ↔ 1), which encourages broader exploration and increases the chance of reaching high-quality solutions.
>
> The key point we emphasize is that **VA was not designed to be effective in isolation**. Rather, its role is to create structurally consistent and interpretable intermediate states, serving as a reliable foundation for the reverse process when combined with CCS. Once CCS is applied, which **selectively activates only high-confidence variables at each step**, the VA framework’s structure-preserving property becomes crucial: **it ensures that CCS operates over a low-noise, semantically meaningful space**. Notably, while CCS also improves performance when applied to the Uniform CM baseline, the gain is significantly amplified when combined with VA. This confirms that these two components are complementary and jointly responsible for the observed performance gains.
>
> Building on these improved initial solutions, the introduction of **Vanilla Refinement** further boosts solution quality. On top of that, the addition of **Objective Guidance (OG)** steers the refinement process by prioritizing variables with both high model confidence and **low surrogate cost**, in a fully **gradient-free manner with negligible runtime overhead**.
>
> In summary, this ablation study highlights the **modular and complementary design** of our method:
> - **VA + CCS** collaboratively ensure structural reliability, enabling better solution space exploration and improved outcomes;
> - **OG** promotes objective alignment effectively and efficiently, resulting in better solutions.
>
> We will include this extended analysis and clarification in the final version to better support the empirical strength of our approach.
>
> ---
>
> >**Supplementary Experiments on a New Problem: Capacitated Vehicle Routing Problem**
>
> At the end of this rebuttal, we would like to present an additional experiment on the Capacitated Vehicle Routing Problem (CVRP), which we believe provides further strong empirical support for our methodology.  **Due to space limitations, detailed results are provided in our response to Reviewer RWGz, Q3.**
>
> **To the best of our knowledge, StruDiCO is the first heatmap-based model to surpass sequence-based methods on CVRP, highlighting the potential of structure-guided global prediction in solving combinatorial optimization problems with complex constraints.**
>
>
> ---
> #### **References**
>
> #### \[1] GLOP: Learning Global Partition and Local Construction for Solving Large-scale Routing Problems in Real-time, AAAI, 2024.
>
> ---
>
> We hope this response could help ease your concern and wish to receive your further feedback soon.
>
> Best regards,
>
> The Authors

---

> ### Author Response · Authors · 2025-08-07
>
> Dear Reviewer 2ZPS,
>
> Thank you again for your thoughtful review and for highlighting both the strengths and the potential of our proposed method.
>
> **As the NeurIPS 2025 author-reviewer discussion phase has been extended**, we would like to kindly follow up in case you had any additional thoughts after reading our rebuttal and supplementary results.
>
> We noticed that the rating has been updated (though hidden from us, as per policy), and we sincerely hope that our detailed responses have addressed your concerns. If there are any remaining issues you'd like to further discuss, or if you would like to briefly comment on whether our clarifications were satisfactory, we would greatly appreciate your feedback.
>
> In particular, we responded in detail to your comments regarding:
>
> + A comprehensive **ablation study** analyzing the contributions of Variable Absorption (VA), Constrained Consistency Sampling (CCS), and Objective-Guided Refinement (OG);
>
> + **Scalability** on TSP-10,000, and comparisons with **DISCO** and GLOP;
>
> + The practical applicability of our method to more constrained problems like **CVRP**, demonstrating **generalization** beyond TSP and MIS.
>
> Please don’t hesitate to let us know if any further clarification would help you finalize your assessment.
>
> Thank you again for your time and consideration.
>
> Best regards,
>
> The Authors

---

### Official Review · Reviewer_XM73 · 2025-06-28

**Clarity:** 3
**Significance:** 2
**Originality:** 2
**Rating:** 4
**Confidence:** 4

**Summary:**

This paper proposes a structured discrete diffusion framework, StruDiCO, tailored for combinatorial optimization (CO) problems. It employs Variable-Absorption as the forward noising strategy during training and Constrained Consistency Sampling for reverse inference during testing. Leveraging the structural consistency throughout the inference process, the authors further introduce a gradient-free refinement method to enhance solution quality. While the paper presents a novel and promising approach, the reviewer has some concerns regarding the completeness of the experimental analysis and the clarity of the objective-aware decoding mechanism.

**Questions:**

1. The proposed Variable-Absorption Diffusion and Constrained Consistency Sampling seem to be the core innovations. However, the experimental section lacks clear ablation studies isolating their contributions. This makes it difficult to assess which component contributes most to the performance gains.
2. The decoding strategy is guided by a multiplicative score between model confidence and task-specific penalties. This formulation seems simplistic and may bias the search toward either confidence or task cost. The authors are encouraged to elaborate on how this trade-off is balanced.
3. The reviewer is surprised that the gradient-free refinement achieves significant quality improvements in only 3–5 steps. Typically, gradient-free methods (e.g., 2-opt) require more iterations to converge. The authors should provide a more detailed explanation or empirical evidence to justify this rapid convergence.
4. Gradient-free methods often scale poorly to large problem instances, which raises concerns from the reviewer about the applicability of StruDiCO in such settings. The authors are encouraged to compare against recent CO solvers. In particular, the reviewer notes that recent work has explored the use of diffusion solvers for tackling large-scale combinatorial optimization problems—such as ‘DISCO: Efficient Diffusion Solver for Large-Scale Combinatorial Optimization Problems’. Comparing StruDiCO with these approaches would further strengthen the contribution of this paper.
5. Given the fast evolution of recent combinatorial optimization research, the reproducibility of results is crucial for future comparison and follow-up work. The reviewer recommends that the authors consider releasing their codebase to strengthen the impact and reproducibility of their contributions.

**Ethical Concerns:**

["NO or VERY MINOR ethics concerns only"]

**Final Justification:**

The reviewer has carefully read the authors' response, including the questions and replies addressed to other reviewers. The authors have provided thorough answers to my concerns. In addition, the performance of StruDiCO on the CVRP task, beyond TSP and MIS, further supports the method's effectiveness. My concerns have been largely addressed, and I maintain my original positive score.

**Limitations:**

This paper provides an insightful approach to designing diffusion solvers for combinatorial optimization. It outperforms recent methods in terms of both solution quality and inference efficiency. However, the reviewer believes that additional ablation studies and clarifications regarding the decoding process would substantially improve the clarity and credibility of the work.

**Paper Formatting Concerns:**

No.

**Quality:**

3

**Strengths And Weaknesses:**

1. The paper proposes a structured diffusion solver with a novel variable-absorption mechanism. It further integrates a gradient-free local refinement strategy that improves performance over recent advances while offering a faster inference advantage.
2. Compared to gradient-based methods, the proposed gradient-free refinement appears easier to implement and more applicable in practice. This increases the potential applicability of StruDiCO.
3. The paper provides clear methodological details and supports the algorithmic design with theoretical analysis.

---

> ### Author Rebuttal · Authors · 2025-07-26
>
> Dear Reviewer XM73,
>
> Thanks for your insightful comments and for acknowledging model design and empirical performance. Below we respond to the specific comments.
>
> > **Q1: Clarifying the individual contributions of Variable-Absorption and Constrained Consistency Sampling via ablation.**
>
> We provide a detailed ablation study to isolate the contribution of each component in our method. The analysis focuses on three parts:
>
> 1. The role of **Variable-Absorption (VA)** mechanism,
> 2. The effect of **Constrained Consistency Sampling (CCS)** in the main inference process, and
> 3. The contribution of the **Objective-Aware Refinement** module applied on the initial solution solved by StruDiCO with $T_s=5$.
>
> Specifically, we compare several variants, including:
> - **Uniform CM**: the uniform cosistency model (CM) proposed in Fast T2T;
> - **VA CM**: replacing the uniform CM with the Variable-Absorption CM;
> - **+CCS**: where low-confidence variables are masked;
> - **Vanilla Refinement**: which refines the initial solution without gradient-free objective guidance (OG).
>
> #### Table: Ablation Study (Gap, % ↓)
>
> | Variant                      | TSP-50 | TSP-100 | TSP-500 | TSP-1000 | MIS-RB | MIS-ER |
> |-----------------------------|--------|---------|---------|----------|--------|--------|
> | Uniform CM (Fast T2T,  \$T\_s\$=5)       | 0.031     | 0.101     | 0.922     | 1.167     | 6.023     | 15.898     |
> | VA CM (StruDiCO)               | 0.032  | 0.163   | 1.091   | 1.183    | 7.476  | 23.461 |
> |Uniform +CCS |     0.025 | 0.081   | 0.898   | 1.074  | 5.878  | 16.522 |
> | VA + CCS (StruDiCO, \$T\_s\$=5)      | 0.019  | 0.067   | 0.728   | 0.797    | 3.899  | 11.450 |
> | Vanilla Refinement          | 0.012  | 0.033   | 0.381   | 0.672    | 1.922  | 7.611  |
> | + OG (StruDiCO, \$T\_s\$=5, \$T\_g\$=5)    | **0.011** | **0.025** | **0.319** | **0.473**  | **1.707** | **5.579** |
>
> As shown, replacing the uniform noise (used in DIFUSCO, T2T, and FastT2T) with the Variable-Absorption (VA) mechanism alone leads to a performance degradation. This is because VA introduces a monotonic absorption process, where variables are only deactivated (i.e., 1 → 0) during the forward diffusion. While this preserves structural continuity at each timestep, it results in overly conservative trajectories when used alone—**limiting exploration and diversity in the reverse process**. In contrast, the Uniform Consistency Model (CM) allows symmetric noise injection (both 0 ↔ 1), which encourages broader exploration and increases the chance of reaching high-quality solutions.
>
> The key point we emphasize is that **VA was not designed to be effective in isolation**. Rather, its role is to create structurally consistent and interpretable intermediate states, serving as a reliable foundation for the reverse process when combined with CCS. Once CCS is applied, which **selectively activates only high-confidence variables at each step**, the VA framework’s structure-preserving property becomes crucial: **it ensures that CCS operates over a low-noise, semantically meaningful space**. Notably, while CCS also improves performance when applied to the Uniform CM baseline, the gain is significantly amplified when combined with VA. This confirms that these two components are complementary and jointly responsible for the observed performance gains.
>
> Building on these improved initial solutions, the introduction of **Vanilla Refinement** further boosts solution quality. On top of that, the addition of **Objective Guidance (OG)** steers the refinement process by prioritizing variables with both high model confidence and **low surrogate cost**, in a fully **gradient-free manner with negligible runtime overhead**.
>
> In summary, this ablation study highlights the **modular and complementary design** of our method:
> - **VA + CCS** collaboratively ensure structural reliability, enabling better solution space exploration and improved outcomes;
> - **OG** promotes objective alignment effectively and efficiently, resulting in better solutions.
>
> We will include this extended analysis and clarification in the final version to better support the empirical strength of our approach.
>
> ---
>
> > **Q2: The decoding strategy is guided by a multiplicative score between model confidence and task-specific penalties. This formulation seems simplistic and may bias the search toward either confidence or task cost.**
>
> Thank you for your constructive comment. To better explore the trade-off between model confidence and task-specific penalties, we extended our original formulation. Specifically, we tested a generalized decoding score defined as: $confidence / (\epsilon + penalty)$,
> where `ε` serves as a balancing term. Notably, our original multiplicative form can be viewed as a special case: `ε = 0` for TSP and `ε = 1` for MIS.
>
> In our new experiments, we found that the TSP task is insensitive to ε due to its relatively smooth objective landscape. However, for MIS, tuning ε led to noticeable improvements. As shown in the table below, setting ε = 10⁻² for RB graphs and ε = 10⁻³ for ER graphs yielded better results than the original ε = 1 setting.
>
> #### Table: Effect of ε on MIS Gap (% ↓)
>
> | ε         | 1e-8  | 1e-5  | 1e-4  | 1e-3  | **1e-2** | 0.1  | **1**   | 10     |
> |-----------|-------|-------|-------|-------|----------|-------|--------|--------|
> | MIS-RB    | 1.673 | 1.644 | 1.650 | 1.636 | **1.596** | 1.692 | 1.707  | 1.709  |
> | MIS-ER    | 5.363 | 5.492 | 5.246 | **4.970** | 5.231  | 5.413 | 5.413  | 5.468  |
>
> These results suggest that our decoding formulation is not inherently biased, but rather flexible and tunable to different CO tasks. We will include these findings and the general formulation in the final version.
>
> ---
>
> > **Q3: Explaining the fast convergence of gradient-free refinement.**
>
> Thank you for the insightful question. Unlike classical local search methods such as 2-opt, which typically apply a **single** local perturbation per step and require thousands of iterations to converge, our refinement mechanism performs a **batch update**: at each step, it simultaneously perturbs all low-confidence variables using variable-absorption noising, followed by selection based on objective-aware decoding. This parallel refinement strategy significantly accelerates convergence.
>
> To support this, we present the following results on TSP-500, showing the trade-off between performance and computation time over different refinement steps:
>
> #### Table: Refinement Step Efficiency (TSP-500)
>
> | Steps     | 0     | 1     | 2     | 3     | 4     | 5     | 6     | 7     | 8     |
> |-----------|-------|-------|-------|-------|-------|-------|-------|-------|-------|
> | Gap (%)   | 0.732 | 0.534 | 0.448 | 0.377 | 0.322 | 0.315 | 0.292 | 0.248 | 0.238 |
> | Time (s)  | 0.255 | 0.339 | 0.415 | 0.491 | 0.569 | 0.640 | 0.724 | 0.823 | 0.902 |
>
> As shown, the refinement process leads to steady improvements over steps. Notably, the most significant quality gains are achieved within the first 3–5 steps, making this range a good trade-off point between performance and efficiency. Although further steps continue to improve the solution, the **marginal gain diminishes**, while the **computational cost increases linearly**.
>
> ---
>
> > **Q4: Scalability of StruDiCO and comparison with large-scale CO solvers (e.g., DISCO).**
>
> To evaluate the scalability of our method, we conducted additional experiments on the large-scale **TSP-10,000** dataset. We compare StruDiCO with diffusion-based baselines, including **DIFUSCO** and **DISCO**, and a divide-and-conquer (DC) approach, **GLOP** [1].
>
> #### Table: Results on TSP-10000
>
> | Method                 | Type         | Length ↓ | Drop ↓ | Time   |
> |------------------------|--------------|----------|--------|--------|
> | LKH   | Heuristics   | 71.77    | 0.00%  | 332s   |
> | GLOP (more revisions)  | DC           | 75.29    | 4.90%  | 15s    |
> | DIFUSCO ($T_s$=100)    | SL+G+2OPT    | 73.91    | 2.98%  | 124s   |
> | DISCO                  | SL+G+2OPT    | 73.84    | 2.88%  | 92s    |
> | **StruDiCO ($T_s$=5)** | SL+G+2OPT    | **72.90**| **1.57%** | **60s** |
>
> It is important to note that as the problem size increases (e.g., 100K nodes), the primary bottleneck during inference is not the diffusion (or consistency) model itself. Rather, **the major computational overhead arises from the greedy decoding procedure**, where edges or nodes with the highest confidence are sequentially inserted into a partial solution until a feasible one is constructed. This step can become prohibitively time-consuming in large-scale instances. We believe that the scalability of StruDiCO could be further improved by combining its backbone generative model with a more efficient decoding strategy or a divide-and-conquer approach. This presents a promising direction for future research.
>
> We will include these large-scale results and comparisons with DISCO in the final version to strengthen the contribution.
>
> ---
>
> > **Q5: Code release.**
>
> We appreciate your suggestion. We are committed to releasing our full codebase and pretrained models upon publication. The objective of this work is to provide reference and contributions for the ML4CO community. Open-sourcing the code is a fundamental obligation we undertake.
>
> ---
>
> >**Supplementary Experiments on a New Problem: Capacitated Vehicle Routing Problem**
>
> At the end of this rebuttal, we present an additional experiment on the CVRP, which we believe would provide additional strong empirical support for our methodology. **Due to space limitations, please refer to our response to Reviewer RWGz, Q3, for the results.**
>
> ---
> #### **References**
>
> #### \[1] GLOP: Learning Global Partition and Local Construction for Solving Large-scale Routing Problems in Real-time, AAAI, 2024.
> ---
> We hope this response could help address your concerns, and we are more than happy to address any further concerns you may have.
>
> Best regards,
>
> The Authors

---

> > ### Comment · Reviewer_XM73 · 2025-08-06
> > **I maintain my original positive score**
> >
> > The reviewer has carefully read the authors' response, including the questions and replies addressed to other reviewers. The authors have provided thorough answers to my concerns. In addition, the performance of StruDiCO on the CVRP task, beyond TSP and MIS, further supports the method's effectiveness. My concerns have been largely addressed, and I maintain my original positive score.

---

> > > ### Author Response · Authors · 2025-08-06
> > >
> > > We truly appreciate your time, effort, and positive evaluation throughout the review process.
> > >
> > > We are pleased to hear that our responses have largely addressed your concerns. We will incorporate all supplementary results and explanations into the final version of the paper. We will also continue to refine our work based on your valuable suggestions.

---

### Official Review · Reviewer_RWGz · 2025-07-03

**Clarity:** 2
**Significance:** 2
**Originality:** 3
**Rating:** 4
**Confidence:** 2

**Summary:**

This paper proposes StruDiCO, a structured denoising framework for combinatorial optimization (CO). Its two main novelties are:
- A structured noising process that enforces monotonicity in the forward and reverse directions, aiming to make the denoising path more efficient.
- A gradient-free decoding approach, with no dependency on the gradients from objectives.
It claims significant improvements in speed (3.5x) and memory saving (70%), and improved solution quality as well

**Questions:**

- The improvements in TSP and MIS looks minimal. Can you elaborate on why the performance gap is small, and under what settings the proposed method provides more substantial benefits?
- Can you provide a more detailed ablation that showcase the contribution of the gradient-free refinement? How much does each component in StruDiCO contribute to the final performance improvement?
- Could StruDiCO be generalized well to more CO problems?

**Ethical Concerns:**

["NO or VERY MINOR ethics concerns only"]

**Final Justification:**

The authors have provided thorough responses to the questions raised during the review, effectively addressing my concerns. As a result, I have updated my rating.

**Limitations:**

yes

**Quality:**

2

**Strengths And Weaknesses:**

Strengths:
- The gradient-free refinement framework with eliminated back-propagation is a practical contribution
- The paper proposes a threshold-based constrained sampling strategy with theoretical grounding

Weaknesses:
- Marginal empirical improvements: while the method is cleanly designed, its gains over strong baselines are often small or inconsistent, making it difficult to justify the additional complexity of introducing the structured process.
- Gradient-free approach is not critical in tested examples: the tested tasks TSP and MIS do not present known issues with gradient use, so it’s unclear when this feature would be critical.
- The results are based on TSP and MIS which are two common tasks in CO. More diverse tasks would show broader applicability of the proposed method

---

> ### Author Rebuttal · Authors · 2025-07-26
>
> Dear Reviewer RWGz,
>
> Thanks for the thorough review and valuable comments. Below we respond to your specific comments.
>
> ---
>
> > **Q1: The improvements in TSP and MIS look minimal. Can you elaborate on why the performance gap is small, and under what settings the proposed method provides more substantial benefits?**
>
> We respectfully disagree with the view that the improvements on TSP and MIS are minimal. Across various settings, **StruDiCO achieves average relative performance gains of 19.6% (TSP) and 38.1% (MIS)** over the strongest diffusion-based baselines (e.g., Fast T2T). While the absolute reductions in optimality gap may appear numerically small, it is important to recognize that these improvements are achieved in an **extremely low-error regime**, where prior methods were already within 1% of optimality. Improving further under such conditions is inherently more challenging and thus more meaningful.
>
> Moreover, **StruDiCO delivers substantial improvements in efficiency**. In particular, StruDiCO introduces a gradient-free refinement mechanism that significantly accelerates inference compared to existing gradient-based refinement methods (e.g., T2T, Fast T2T). Specifically, we observe up to 70\% reduction in GPU memory usage and a 3.5$\times$ speedup in inference time, as shown in Tables 2–5 and Fig. 2 of the main paper. These results demonstrate that StruDiCO advances both solution quality and computational efficiency, offering a practical and scalable alternative to gradient-based refinement.
>
> Beyond performance, **StruDiCO also enhances interpretability** by maintaining structurally valid intermediate states throughout the diffusion process. This structure-preserving design facilitates step-wise understanding of the solution construction, enabling deeper analysis of combinatorial patterns—an aspect particularly valuable for operations research and scientific decision-making applications.
>
>  Please kindly note that reviewers Z5NJ, XM73, and 2ZPS all acknowledge the empirical results of our method.
>
> As for the settings under which our method offers more substantial benefits, we observe that the improvements are especially **pronounced in larger-scale and structurally complex problem instances**—such as **TSP-1000** and **MIS-ER** (see Fig. 2). In these challenging scenarios, the advantages of structure-preserving diffusion and gradient-free objective-aware refinement become more evident, leading to larger gains in both solution quality and efficiency.
>
> ---
>
> > **Q2: Can you provide a more detailed ablation that showcases the contribution of the gradient-free refinement? How much does each component in StruDiCO contribute to the final performance improvement?**
>
> Thank you for the thoughtful question. We provide a detailed ablation study to isolate the contribution of each component in our method. The analysis focuses on two parts:
>
> 1. The effect of **Constrained Consistency Sampling (CCS)** in the main inference process, and
> 2. The contribution of the **gradient-free objective-aware refinement** module applied on the initial solution solved by StruDiCO with $T_s=5$.
>
> Specifically, we compare several variants, including:
> - **VA only**: using the Variable-Absorption diffusion with greedy decoding;
> - **+CCS**: where low-confidence variables are masked;
> - **Vanilla Refinement**: which refines the initial solution without gradient-free objective guidance (OG).
>
> #### Table: Ablation Study (Gap, % ↓)
>
> | Variant                      | TSP-50 | TSP-100 | TSP-500 | TSP-1000 | MIS-RB | MIS-ER |
> |-----------------------------|--------|---------|---------|----------|--------|--------|
> | VA only                     | 0.032  | 0.163   | 1.091   | 1.183    | 7.476  | 23.461 |
> | + CCS (StruDiCO, \$T\_s\$=5)      | 0.019  | 0.067   | 0.728   | 0.797    | 3.899  | 11.450 |
> | Vanilla Refinement          | 0.012  | 0.033   | 0.381   | 0.672    | 1.922  | 7.611  |
> | + OG (StruDiCO, \$T\_s\$=\$T\_g\$=5)    | **0.011** | **0.025** | **0.319** | **0.473**  | **1.707** | **5.579** |
>
> From the results, we first observe that incorporating **Constrained Consistency Sampling (CCS)** significantly enhances the performance of Variable Absorption (VA), yielding up to a **51% reduction in optimality gap** (e.g., MIS-ER: 23.461 → 11.450). This improvement arises because CCS prevents low-confidence variables from being prematurely activated, resulting in more stable and reliable inference trajectories.
>
> Building on these improved initial solutions, the introduction of **Vanilla Refinement** further boosts solution quality. On top of that, the addition of **Objective Guidance (OG)** steers the refinement process by prioritizing variables with both high model confidence and low surrogate cost—in a fully **gradient-free manner with negligible runtime overhead**.
> For example: TSP-500 improves from 0.381 → 0.319 ; MIS-ER improves from 7.611 → 5.579
>
>
> In summary, this ablation study highlights the **modular and complementary design** of our method:
> - **VA + CCS** collaboratively ensure structural reliability, enabling better solution space exploration and improved outcomes;
> - **OG** promotes objective alignment effectively and efficiently, resulting in better solutions.
>
> Owing to space limitations, this response primarily focuses on clarifying the contribution to performance improvements. For a **more comprehensive ablation analysis**, please refer to our response to Reviewer XM73, Q1. In addition, Fig. 2 in the main paper provides a detailed comparison with **gradient-based refinement** methods, demonstrating that our gradient-free refinement consistently outperforms them in both solution quality and computational efficiency.
>
>
> ---
>
> > **Q3: Could StruDiCO be generalized well to more CO problems?**
>
> In fact, StruDiCO can handle a wide range of constrained combinatorial optimization (CO) problems. Leveraging the expressive capacity of generative models, our approach learns to approximate the underlying constraint structure, producing heatmaps that are already close to feasible. We then apply lightweight heuristic post-processing (e.g., greedy decoding) to convert these outputs into strictly feasible solutions. This paradigm enables general applicability to arbitrary constraints and aligns with the inference strategies adopted in prior representative works such as Diffusco, T2T, and Fast T2T.
>
> To further demonstrate the generality of StruDiCO beyond classical edge-selection (TSP) and node-selection (MIS) tasks, we extend it to the **Capacitated Vehicle Routing Problem (CVRP)**—a more challenging CO task that involves both global routing structures and hard feasibility constraints, such as customer demands and vehicle capacities. The **refinement procedure for CVRP follows the same design as that for TSP**: greedy decoding is performed based on the learned heatmap, with the objective guided by a distance-based penalty term. We evaluate StruDiCO on CVRP-50, CVRP-100, and CVRP-200 under standard benchmark settings \[1]. To further improve both solution feasibility and cost, we incorporate a local search heuristic (Classic-LS \[2]) during inference.
>
> **To the best of our knowledge, StruDiCO is the first heatmap-based model that surpasses sequence-based models on VRPs.**
>
>
> | Algorithm                            | Type       |  Length↓ | Drop↓     | CVRP 50 Time  |  Length↓ | Drop↓     | CVRP 100 Time  |  Length↓ |  Drop↓     | CVRP 200 Time  |
> | ------------------------------------ | ---------- | --------------- | --------- | ----- | ---------------- | --------- | ----- | ---------------- | --------- | ----- |
> | HGS \[3]                                 | Heuristics | 10.37           | 0.00%     | 1s    | 15.56            | 0.00%     | 20s   | 19.63            | 0.00%     | 60s   |
> | Sym-NCO \[1]                             | RL+LS      | 10.57           | 1.95%     | 0.09s | 15.93            | 2.37%     | 0.19s | 20.19            | 2.86%     | 0.36s |
> | StruDiCO (\$T\_s\$=3,\$T\_g\$=3) | SL+LS      | 10.48       | 1.12% | 0.05s | 15.85        | 1.88% | 0.11s | 20.25        | 3.19% | 0.26s |
> | **StruDiCO (\$T\_s\$=5,\$T\_g\$=5)** | SL+LS      | **10.45**       | **0.85%** | 0.07s | **15.80**        | **1.53%** | 0.17s | **20.16**        | **2.71%** | 0.38s |
>
> ---
>
> > **W2: Gradient-free approach is not critical in tested examples: the tested tasks TSP and MIS do not present known issues with gradient use, so it’s unclear when this feature would be critical.**
>
> We respectfully clarify that although TSP and MIS are inherently discrete and non-differentiable problems, recent diffusion-based solvers, such as **T2T and FastT2T**, still **rely on gradient-based refinement** over relaxed continuous representations to improve solution quality. In contrast, our method achieves superior performance without any gradient computation, demonstrating that a gradient-free refinement framework can be both effective and efficient. This not only avoids the complexity and overhead associated with backpropagation through relaxed spaces but also enhances the method’s applicability to a broader range of combinatorial problems where gradient-based refinement may be infeasible or suboptimal.
>
> ---
>
> #### **References**
> #### \[1] RL4CO: An Extensive Reinforcement Learning for Combinatorial Optimization Benchmark, 2024
>
> #### \[2] A Simple and Effective Evolutionary Algorithm for the Vehicle Routing Problem, 2004
>
> #### \[3] A Hybrid Genetic Algorithm for Multidepot and Periodic Vehicle Routing Problems, 2012
>
> ---
>
> We hope our point-by-point clarificaitons and supplementary experiments have satisfactorily addressed your concerns. We remain fully committed to involving further discussions with you towards an elevated evaluation of our work.
>
> Best regards,
>
> The Authors

---

> > ### Comment · Reviewer_RWGz · 2025-08-08
> >
> > I sincerely appreciate the authors' efforts in addressing all concerns, clarifying misunderstandings, and conducting additional experiments. I will update my rating accordingly.

---

> > > ### Author Response · Authors · 2025-08-08
> > >
> > > Thank you for your thoughtful response and acknowledgment. We are glad to hear that our clarifications and additional results have addressed your concerns. We sincerely appreciate your constructive feedback and will reflect your suggestions in the final version of the paper.

---

> ### Author Response · Authors · 2025-08-07
>
> Dear Reviewer RWGz,
>
> Thank you again for your review and thoughtful comments.
>
> **As the NeurIPS 2025 program chairs have extended the author-reviewer discussion phase**, we would like to kindly follow up to ensure you’ve had a chance to revisit our detailed rebuttal and supplementary results. We would greatly appreciate your thoughts or clarifications if any part of our response warrants further discussion.
>
> In particular, we have carefully addressed the key concerns you raised:
>
> + **On the perceived marginal improvements**: We respectfully clarified that StruDiCO achieves average relative gains of 19.6% (TSP) and 38.1% (MIS) over strong diffusion-based baselines (e.g., Fast T2T), in an already low-error regime. We also showed that these gains are more pronounced on larger-scale or structurally complex instances, such as TSP-1000 and MIS-ER.
>
> + **On the contribution of gradient-free refinement**: We provided a detailed ablation study, which demonstrated the complementary effects of Variable Absorption (VA), Constrained Consistency Sampling (CCS), and Objective-Guided Refinement (OG), as well as the efficiency and effectiveness of our fully gradient-free approach.
>
> + **On generalizability beyond TSP and MIS**: We extended StruDiCO to the Capacitated Vehicle Routing Problem (CVRP), showing that it not only handles more complex constraints but also outperforms sequence-based models on standard CVRP benchmarks.
>
> + **On the necessity of gradient-free refinement**: We clarified that while TSP and MIS are discrete tasks, recent diffusion solvers (e.g., T2T, Fast T2T) still rely on gradient-based refinement over continuous relaxations. In contrast, StruDiCO offers a practical and scalable alternative without requiring gradient computations.
>
> We sincerely value your feedback and would be grateful if you could let us know whether these responses help clarify your concerns, or if there are additional points you’d like to discuss. Your input would be invaluable in helping us further improve and present our work as clearly and rigorously as possible.
>
> Thank you again for your time and effort in reviewing our submission.
>
> Best regards,
>
> The Authors

---

### Official Review · Reviewer_Z5NJ · 2025-07-08

**Clarity:** 2
**Significance:** 3
**Originality:** 2
**Rating:** 4
**Confidence:** 3

**Summary:**

The paper introduces StruDiCO, a structured denoising diffusion framework designed to solve combinatorial optimization (CO) problems efficiently and interpretably. StruDiCO uses Structured Forward Diffusion to ensure that every intermediate state remains a structurally valid partial solution. This “structure-preserving” approach allows the model to maintain interpretability and stability across the denoising trajectory. In the reverse process, StruDiCO selectively includes variables based on a confidence threshold, avoiding low-confidence predictions. In addition, StruDiCO uses an objective-aware refinement framework that efficiently incorporates
 task objectives without requiring backward gradient computation. Empirical results on standard benchmarks - including the Traveling Salesman Problem (TSP) and Maximum Independent Set (MIS) - demonstrate that StruDiCO outperforms prior diffusion models in both solution quality and inference speed, while using significantly less memory.

**Questions:**

- The authors can provide a more detailed explanation of why their method outperforms T2T and DIFUSCO in terms of accuracy.

- How much labeled data is enough for your method?

**Ethical Concerns:**

["NO or VERY MINOR ethics concerns only"]

**Limitations:**

Yes, the paper addressed the limitations at Appendix G

**Quality:**

3

**Strengths And Weaknesses:**

Strengths:

- The paper proposes a new solution for CO to bridge the gap between step-wise interpretability and global distribution modeling. Their solution, StruDiCO introduces a variable absorption noising process that gradually deactivates variables during the forward diffusion stage. Unlike prior methods (e.g., T2T, DIFUSCO) that apply random or uniform noise, this approach ensures that each intermediate state remains a structurally valid partial solution. In the reverse process, StruDiCO only selects variables whose predicted confidence exceeds a set threshold (δ), discarding low-confidence choices.

- StruDiCO achieves a significantly faster inference and lower memory usage than prior diffusion-based models.

Weaknesses:

- It lacks a valid explanation on why their method is empirically strong compared to other diffusion-based methods

The refinement step in StruDiCO is based on heuristics. It is unclear how their method is efficient for the new CO problem

- Like other diffusion-based methods, their solution needs a substantial number of labeled optimal or near-optimal solutions.

---

> ### Author Rebuttal · Authors · 2025-07-26
>
> Dear Reviewer Z5NJ,
>
> Thanks for your recognition and valuable questions.
>
> ---
>
> > **Q1: Clarification on why the proposed method achieves higher accuracy than other diffusion-based methods.**
>
> To better explain the performance gain of our method over existing diffusion-based approaches (e.g., those using uniform diffusion), we performed an ablation study to isolate the individual contributions of Variable Absorption (VA) and Constrained Consistency Sampling (CCS).
>
> **Ablation Study (Gap, %↓):**
>
> | Variants          | TSP 50    | TSP 100   | TSP 500   | TSP 1000  | MIS RB    | MIS ER     |
> | ----------------- | --------- | --------- | --------- | --------- | --------- | ---------- |
> | Uniform CM        | 0.031     | 0.101     | 0.922     | 1.167     | 6.023     | 15.898     |
> | VA CM             | 0.032     | 0.163     | 1.091     | 1.183     | 7.476     | 23.461     |
> |Uniform+CCS |     0.025 | 0.081   | 0.898   | 1.074  | 5.878  | 16.522 |
> | **VA+CCS (Ours)** | **0.019** | **0.067** | **0.728** | **0.797** | **3.899** | **11.450** |
>
> As shown, replacing the uniform noise (used in DIFUSCO, T2T, and FastT2T) with the Variable-Absorption (VA) mechanism alone leads to a performance degradation. This is because VA introduces a monotonic absorption process, where variables are only deactivated (i.e., 1 → 0) during the forward diffusion. While this preserves structural continuity at each timestep, it results in overly conservative trajectories when used alone, **limiting exploration and diversity in the reverse process**. In contrast, the Uniform Consistency Model (CM) allows symmetric noise injection (both 0 ↔ 1), which encourages broader exploration and increases the chance of reaching high-quality solutions.
>
> The key point we emphasize is that **VA was not designed to be effective in isolation**. Rather, its role is to create structurally consistent and interpretable intermediate states, serving as a reliable foundation for the reverse process when combined with CCS. Once CCS is applied, which **selectively activates only high-confidence variables at each step**, the VA framework’s structure-preserving property becomes crucial: **it ensures that CCS operates over a low-noise, semantically meaningful space**. Notably, while CCS also improves performance when applied to the Uniform CM baseline, the gain is significantly amplified when combined with VA.  This confirms that these two components are complementary and jointly responsible for the observed performance gains.
>
> This response primarily addresses your concern regarding the **impact of diffusion model variations on accuracy**. For a more comprehensive ablation analysis, please refer to our response to Reviewer XM73, Q1.
>
> ---
>
> > **Q2: How much labeled data is enough for your method?**
>
> For fairness, we adopt the **same** training set sizes as those used in DIFUSCO and FastT2T.
>
>
> | Problem  | # Train Instances |
> | -------- | ----------------- |
> | TSP-50   | 1.58M             |
> | TSP-100  | 1.58M             |
> | TSP-500  | 128K              |
> | TSP-1000 | 64K               |
> | MIS-RB   | 90K               |
> | MIS-ER   | 163K              |
>
> ---
>
> > **W2: The refinement step in StruDiCO is based on heuristics. It is unclear how their method is efficient for the new CO problem.**
>
> In fact, StruDiCO can handle a wide range of constrained combinatorial optimization (CO) problems. Leveraging the expressive capacity of generative models, our approach learns to approximate the underlying constraint structure, producing heatmaps that are already close to feasible. We then apply lightweight heuristic post-processing (e.g., greedy decoding) to convert these outputs into strictly feasible solutions. This paradigm enables general applicability to arbitrary constraints and aligns with the inference strategies adopted in prior representative works such as Diffusco, T2T, and Fast T2T.
>
> To further demonstrate the generality of StruDiCO beyond classical edge-selection (TSP) and node-selection (MIS) tasks, we extend it to the **Capacitated Vehicle Routing Problem (CVRP)**—a more challenging CO task that involves both global routing structures and hard feasibility constraints, such as customer demands and vehicle capacities. The **refinement procedure for CVRP follows the same design as that for TSP**: greedy decoding is performed based on the learned heatmap, with the objective guided by a distance-based penalty term. We evaluate StruDiCO on CVRP-50, CVRP-100, and CVRP-200 under standard benchmark settings \[1]. To further improve both solution feasibility and cost, we incorporate a local search heuristic (Classic-LS \[2]) during inference.
>
> **To the best of our knowledge, StruDiCO is the first heatmap-based model that surpasses sequence-based models on VRPs.**
>
>
> | Algorithm                            | Type       |  Length↓ | Drop↓     | CVRP 50 Time  |  Length↓ | Drop↓     | CVRP 100 Time  |  Length↓ |  Drop↓     | CVRP 200 Time  |
> | ------------------------------------ | ---------- | --------------- | --------- | ----- | ---------------- | --------- | ----- | ---------------- | --------- | ----- |
> | HGS \[3]                                 | Heuristics | 10.37           | 0.00%     | 1s    | 15.56            | 0.00%     | 20s   | 19.63            | 0.00%     | 60s   |
> | Sym-NCO \[1]                             | RL+LS      | 10.57           | 1.95%     | 0.09s | 15.93            | 2.37%     | 0.19s | 20.19            | 2.86%     | 0.36s |
> | StruDiCO ($T_s=3$,$T\_g=3$) | SL+LS      | 10.48       | 1.12% | 0.05s | 15.85        | 1.88% | 0.11s | 20.25        | 3.19% | 0.26s |
> | StruDiCO ($T\_s=5$,$T\_g=5$) | SL+LS      | **10.45**       | **0.85%** | 0.07s | **15.80**        | **1.53%** | 0.17s | **20.16**        | **2.71%** | 0.38s |
>
> ---
>
> #### **References**
> #### \[1] RL4CO: An Extensive Reinforcement Learning for Combinatorial Optimization Benchmark, 2024
>
> #### \[2] A Simple and Effective Evolutionary Algorithm for the Vehicle Routing Problem, 2004
>
> #### \[3] A Hybrid Genetic Algorithm for Multidepot and Periodic Vehicle Routing Problems, 2012
>
> ---
>
> We hope this response could help ease your concern and wish to receive your further feedback soon.
>
> Best regards,
>
> The Authors

---

> ### Author Response · Authors · 2025-08-07
>
> Dear Reviewer Z5NJ,
>
> Thank you again for your initial review and thoughtful comments.
>
> **As the author-reviewer discussion phase has been extended by 48 hours**, we would like to kindly follow up to see if you had a chance to revisit our rebuttal and supplementary results. We have responded in detail to all your concerns, including:
>
> + A thorough **ablation analysis** addressing why our method outperforms T2T and DIFUSCO in terms of accuracy (Q1);
>
> + Clarification on **data requirements**, aligned with existing diffusion-based solvers (Q2);
>
> + Additional experiments on **CVRP** demonstrating the **generality** of StruDiCO across constrained CO tasks (W2).
>
> Your feedback was instrumental in helping us refine our explanations and highlight the complementary role of Variable Absorption and Constrained Consistency Sampling. We would greatly appreciate it if you could kindly take a moment to let us know whether our responses have addressed your concerns, or if you have any further suggestions or questions.
>
> Thank you once again for your time and support in the review process. We sincerely value your input.
>
> Best regards,
>
> The Authors

---

### Author Response · Authors · 2025-08-06
**General Response**

Dear AC and Reviewers,

We sincerely thank the AC and reviewers for the time and effort in reviewing our paper. Overall, all reviewers recognized the **novelty and practical value of StruDiCO**, highlighting its structured design (Z5NJ, 2ZPS), confidence-guided inference(Z5NJ), and gradient-free refinement (RWGz, XM73, 2ZPS). The method is considered **efficient, interpretable, and easy to implement, with strong theoretical grounding** (RWGz, XM73) and **clear methodology** (XM73). Empirical results on TSP and MIS show **superior solution quality, faster inference, and lower memory usage** compared to prior diffusion models (Z5NJ, RWGz, XM73, 2ZPS), confirming StruDiCO’s effectiveness in neural combinatorial optimization.

While the reviewers overall appreciated the novelty and practical contributions of StruDiCO, they raised several constructive concerns, all of which we have carefully addressed through detailed rebuttals, additional experiments, and clarifications.

+ A common suggestion *(Z5NJ, RWGz, XM73, 2ZPS)* was the need for **clearer ablation studies** to isolate the contributions of the proposed components. We have responded with comprehensive ablations across TSP and MIS, demonstrating the complementary effects of Variable Absorption (VA), Constrained Consistency Sampling (CCS), and Objective-Guided Refinement (OG).

+ We would like to respectfully clarify **two misunderstandings raised by Reviewer RWGz**. First, the perceived marginal gains are in fact substantial relative improvements, **19.6% for TSP and 38.1% for MIS**, achieved over strong baselines such as Fast T2T in an already low-error regime. Second, while TSP and MIS are inherently discrete problems, **recent diffusion solvers like T2T and Fast T2T still rely on gradient-based refinement** over relaxed continuous representations. In contrast, StruDiCO achieves superior performance without any gradient computation, resulting in a 3.5× speedup and 70% memory reduction. This demonstrates that our gradient-free refinement is both efficient and highly effective, offering a scalable alternative to gradient-based solvers.

+ Regarding **generalizability to broader CO tasks** *(Z5NJ, RWGz)*, we extended StruDiCO to the Capacitated Vehicle Routing Problem (**CVRP**), which involves complex global constraints. Our model is the first heatmap-based method to outperform sequence-based models on standard CVRP benchmarks.

+ To address concerns about **scalability to large instances** *(XM73, 2ZPS)*, we evaluated StruDiCO on TSP-10,000, and demonstrated that it outperforms other diffusion-based models, including DISCO, in both solution quality and runtime. We also clarified that the primary bottleneck lies in the greedy decoding procedure, not the diffusion model itself, and suggested directions for further acceleration.

+ In response to concerns about **the formulation and trade-off mechanism of the decoding score** *(XM73)*, we introduced a tunable balancing parameter ε to flexibly control the trade-off between model confidence and objective cost. Experiments showed that performance can be further improved by adjusting ε, particularly on MIS.

+ We also clarified **the fast convergence of our gradient-free refinement** *(XM73)* through analysis of batch-wise parallel updates. Empirical results show that significant improvements are achieved within just 3–5 steps, with diminishing returns thereafter.

+ Finally, addressing **reproducibility and transparency** *(XM73)*, we commit to releasing the full codebase and pretrained models upon publication.

**As the Author-Reviewer discussion phase draws to a close, we remain committed to addressing any remaining concerns and welcome further discussion that may help improve the clarity and impact of our contribution.**



Best regards,

The authors of Paper 7893

---

### Note · Authors · 2025-08-14

Dear AC and Reviewers,

We sincerely thank the reviewers for their time and effort in reviewing our paper and engaging in the discussion. We also appreciate the AC, SAC, and PC for their diligent efforts in organizing and overseeing the review process. We are encouraged to see that during the discussion phase, reviewers RWGz and XM73 stated that our clarifications and supplementary experiments have successfully addressed their concerns, and that the overall reception of our work is positive. All initial concerns, including those related to ablation studies, scalability, and generalization, have been thoroughly addressed in the rebuttal. We will ensure that these clarifications and additional results are clearly incorporated into the final version of the paper.

Best regards,

The authors

---

### Decision · Program_Chairs · 2025-09-17

**Decision:**

Accept (poster)

**Comment:**

The paper proposes a structured denoising diffusion model for incrementally decoding solutions for combinatorial optimization problems, here MIS and TSP.
Reviewers note improvements upon Difusco, being faster, theoretically grounded sampling, improved performance, nice idea. On the other hand, marginal improvements are noted, lacking explanations, ablations, limited experiments.
Given the reviewer consensus, nice theoretical contribution and experimental results, the paper merits acceptance at NeurIPS.